



**A New Characterization of the Upper Waters of the central Gulf of México**
**based on Water Mass Hydrographic and Biogeochemical Characteristics**
*Authors:*
**Gabriela Yareli Cervantes-Diaz[1], Jose Martín Hernández-Ayón[1,5], Alberto Zirino[6],**
**Sharon Zinah Herzka[7], Victor Camacho-Ibar[5], Ivonne Montes[4], Joël Sudre,[3], Juan**
**Antonio Delgado[1,2]**
Author affiliations:
*[1]Facultad de Ciencias Marinas, Universidad Autónoma de Baja California, Transpeninsular*
*Tijuana-Ensenada, no. 3917, Fraccionamiento Playitas, CP 22860. Ensenada, Baja*
*California, México.*
*[2]Instituto Tecnológico de Guaymas/ Tec. Nacional de México, Guaymas, Sonora, México.*
*[3]LEGOS, CNRS/IRD/UPS/CNES UMR 5566, 18 av. Ed Belin, 31401 Toulouse Cedex 9,*
*France*
*[4]Insitituto Geofisico del Perú. Lima, Perú.*
*[5]Instituto de Investigaciones Oceanológicas, Universidad Autónoma de Baja California,*
*Transpeninsular Tijuana-Ensenada, no. 3917, Fraccionamiento Playitas, CP 22860.*
*Ensenada, Baja California, México.*
*[6]Scripps Institution of Oceanography, University of California, San Diego, 9500 Gilman*
*Drive, La Jolla, California 92093, USA.*
*[7]Departamento de Oceanografía Biológica, Centro de Investigación Científica y de*
*Educación Superior de Ensenada (CICESE), Baja California, Carretera Ensenada-Tijuana*
*No. 3918, Zona Playitas, 22860 Ensenada, Baja California, México.*
*Corresponding author
*Instituto de Investigaciones Oceanológicas*
*Universidad Autonoma de Baja California*
*jmartin@uabc.edu.mx;*
**Key Points:**
Gulf of Mexico, Water masses, oxygen dissolved, biogeochemistry.



**Abstract.**
In the Gulf of Mexico (GoM) at least three near-surface water masses are affected by
mesoscale processes that modulate the biogeochemical cycles. Prior studies have presented
different classifications of water masses where the greater emphasis was on deep waters and
not on the surface waters ($\sigma_\theta < 26$ kg·m-3), as in this work. Here presents a new classification
of water masses in the GoM, based on thermohaline properties and dissolved oxygen (DO)
concentration using data from a total of five summer and winter cruises carried out primarily
in the central GoM. The reclassification includes an adjustment to the spatial range of
Caribbean Surface Water (CSW), which is detected only during the summer. This water mass
extends from the surface to H 90 m and features warm waters (27-32 ºC), high salinities (up
to ~36.8), non-detectable nitrate concentration, and negative values of the apparent oxygen
utilization (AOU) of H -27 μmol·kg-1. Below the CSW, the deeper Gulf Common Water
(GCW) was also redefined and characterized by a subsurface DO maximum, with values H
50 μmol·kg-1 higher than that found in surface waters. In winter, a replacement of the CSW
by the GCW affected the biogeochemical composition of surface water as observed from an
increase in nitrate concentrations, positives values of AOU (H 90 μmol·kg-1) and a decrease
in surface temperatures (< 27 ºC). This is because during winter, the Tropical Atlantic Central
Water (TACW) that lies below the GCW is closer to the surface and contributes nutrients
and low DO via strong vertical mixing induced by the windy "Nortes" season. CARS2009
analysis supports the formation of the subsurface maximum of DO during the summer and
disappears in winter. In this work also named surface water that is characterized by a low salt
content (H 33.1) from 0 to 20 m as Freshwater Influenced Surface Water (FISW).





## 1. Introduction.


Circulation in the central Gulf of Mexico's (GoM) is dominated by the Loop Current (LC)
and its associated eddies. Anticyclonic Loop Current Eddies (LCE) H 200 - 300 km in
diameter separate from the LC every 4 to 18 months (Sturges and Leben, 2000; Hall and
Leben, 2016). Another feature associated with the LC is the separation of relatively smaller
cyclonic and anticyclonic eddies throughout the basin, which interact in an apparently
turbulent manner (Schmitz, 2005; Hamilton, 2007a). These eddies extend vertically from a
few hundred to about a thousand meters and appreciably influence the surface dynamics by
modifying the circulation of the GoM (Morey *et al*., 2003a). The position of the LC within
the gulf is variable, and the level of intrusion into the northeastern GoM varies temporally
and spatially (Bunge *et al*., 2002; Delgado J. A. *et al*., 2019).

Near the surface, the spatio-temporal variability in temperature, salinity and dissolved
oxygen (DO) reflect the LC, LCE and other eddy dynamics, freshwater inputs from river
discharge, and seasonal processes such as heat fluxes, evaporation and wind stress that
influence the depth of the mixed layer (Morey *et al*., 2003b; Müller-Karger *et al*., 2015;
Portela *et al*., 2018; Damien *et al*., 2018). A major source of variability in the northern GoM
is the Mississippi River flow, which has been shown to influence areas hundreds of
kilometers from its discharge zone (Morey *et al*., 2003a) over the first 50 m of the water
column (Jochens & DiMarco, 2008; Portela *et al*., 2018). Together, the aforementioned
mechanisms influence water mass characteristics in approximately the first 250 m (or more)
of the water column. For example, upon entering the GoM, the Caribbean Surface Water
(CSW) affects salinity, temperature, and density with values of 34.5 to 36.6; T $\geq$ 25 ºC, and



$\sigma_\theta \leq 24.5$ kg·m-3 (Carrillo *et al*., 2016). Below the CSW, North Atlantic Subtropical
Underwater (NASUW, hereinafter referred to as SUW) can be identified by a salinity
between 36.5 to 36.9 at H 100 to 150 m (Herrig, 2010; Hamilton *et al*., 2018). The Gulf
Common Water (GCW) is distinguished by the relatively homogeneous vertical distribution
of its thermohaline properties, with salinity ranging from 36.3 to 36.49 (Elliott, 1982; Merrell
and Morrison, 1981). Underneath the SUW and GCW, Tropical Atlantic Central Water
(TACW) is found between 300 and 600 m, and is characterized by a DO minimum of 2.3
ml·L-1, T from 7.9 to 20 °C, S from 34.9 to 36.6, and $\sigma_\theta$ from 26.25 to 27.2 kg·m-3 (Vidal *et*
*al*., 1994; Gallegos, 1996; Carrillo *et al*., 2016; Portela *et al*., 2018). The main sources of
variability in the physical and chemical properties of the surface to approx. 250 m (above 26
kg·m-3) can be related to changes in the relative proportions of water masses.

There have been limited surveys of the hydrographic characteristics of the central GoM and
Yucatan Channel within Mexico's Exclusive Economic Zone (including the Campeche Basin
(CB)) based on *in situ* data, and of those, several have been limited to relatively small regions:
For example, Morrison *et al*. (1983) studied the distribution of physical-chemical properties
of the water masses (GCW, TACW, Antarctic Intermediate Water (AAIW) and the mixture
of Caribbean Intermediate Water (CIW) and North Atlantic Deep Water (NADW) and the
NADW) in the northwestern GoM during winter. Similarly, Vidal *et al*. (1994) also
investigated the spatial distribution of thermohaline properties and DO of the GCW, SUW,
TACW, AAIW, as well as the mixture between CIW and NADW, and NADW in the western
region of the GoM during winter and spring. Among these efforts, Rivas *et al*. (2005) studied
the area of the Yucatan Channel, they found five different water masses (SUW, 18º SSW,





TACW, AAIW y NADW). Finally, Hamilton *et al.* (2018) performed an analysis with high-
resolution data from the deeper waters (SUW, AAIW y NADW) of the western and eastern
in the GoM, with results that were consistent with the findings of the previous authors.
Obviously, different water masses may be present depending on the region of the GoM that
is being studied.

In particular, the above authors focused on the role of the dominant LCE's on the
hydrographic characteristics of the central and western GoM (Fig. 1a). However, their
proposed classification did not include near-surface waters; for example, lower salinities
(likely due to river inputs) were not included. Excluding water masses with lower salinities
in the classification scheme limits the inferences that can be made regarding source waters.
This points to the necessity of generating a more detailed classification system in the surface
layers above the 26 kg·m-3 isopycnals, which includes the full range of thermohaline
properties of water masses. When DO concentrations are added to the $\Theta$-$S_A$ diagram as a third
axis, it can be observed that DO shows a high variability (> 200 µmol·kg-1) upwards of the
26 kg·m-3 isopycnal (Fig. 1b). This change in DO is a result of biogeochemical processes via,
photosynthesis, respiration, and exchange with the atmosphere, which also lead to changes
in dissolved inorganic carbon (DIC) and nutrients.

From a biogeochemical perspective, the surface waters of the deep GoM are considered
oligotrophic as they are relatively isolated from the more eutrophic waters of the coast and
continental shelves (Heileman and Rabalais, 2009; Damien *et al*., 2018; but see Martínez-
López y Zavala-Hidalgo, 2009). Near the surface, and far from the coast, low rates of primary
production (low than 0.15 g C m-2 d-1; Biggs y Ressler, 2001) have been reported; however,





productivity in subsurface waters maybe two to three times higher (El-Sayed, 1972; Biggs
and Ressler, 2001). Dynamic features such as mesoscale processes, river inputs, the extent
of the seasonal LC incursion, and wind stress can greatly alter the distribution of chemical
properties in the GoM (Linacre *et al.*, 2015; Damien *et al.*, 2018). Overall, the effect that
water masses have on the seasonal extension of the mixed layer is not well understood,
though its deepening and shallowing play an important role in the rates of primary production
(Damien *et al.*, 2018).

In this work propose a classification for water masses lighter than 26 kg·m$^{-3}$ that more
precisely defines the ranges of thermohaline circulation and DO of the CSW and GCW,
thereby providing a better basis for understanding the processes associated with water mass
formation, distribution, and biogeochemistry in surface waters of the central and western
regions of the GoM. Our purpose is to provide a better tool for studying the drivers that
modulate water mass distribution and its formation in surface waters, as well as the links
between water masses and their biogeochemical properties. The reclassification includes an
adjustment of the thermohaline range of CSW and the GCW. In this work also propose the
formal recognition of Freshwater Influenced Surface Water (FISW) that is characterized by
riverine influence. Finally, examine the role of CSW in the biogeochemistry of the GoM by
comparing the seasonal variations in $T_\theta$ and S in our *in situ* water to the climatological
database CARS 2009.


**2. Data and Methods**



## 2. 1. Data collection

Five oceanographic cruises covering the central region of Mexico's Exclusive Economic Zone were carried out in November 2010, July 2011, February-March 2013, August-September 2015, and July 2016 (XIXIMI-01 through XIXIMI-05, respectively) on board the *R/V Justo Sierra* (Fig. 1c). During these campaigns, a minimum of 30 and maximum of 51 stations per cruise were occupied, and a total of 235 hydrographic casts were performed to characterize the vertical distribution of potential temperature ($T_\theta$), salinity (S), potential density ($\sigma_\theta$), and DO. An SBE 911plus CTD was used; the instrument and sensors were serviced and calibrated regularly.

In addition to CTD casts, water samples were collected for measurements of Dissolved Inorganic Carbon (DIC), nutrients, and DO analyses in 10 or 20 L Niskin bottles at 12 set depths between the surface and bottom. The protocols and best practices established by Dickson *et al.* (2007) were followed for DIC sample collection. For the collection of nutrient samples, 50 ml of seawater were filtered through Whatman GF/F filters previously calcinated at 450 °C for 2 hours, transferred to centrifuge tubes and frozen. Each sample was transported frozen to the laboratory for later analysis. During each cruise, seawater was also routinely sampled for DO (evaluated by the Winkler method) measurements and to calibrate the CTD data. Additionally, the apparent oxygen utilization (AOU) was calculated from DO, T, and S using TEOS-2010 equations. AOU is defined as the deviation of the measured dissolved oxygen from a DO concentration in equilibrium with the atmosphere (Benson and Krause, 1984). When calculating the AOU the DO is corrected for temperature. This allowed us to determine if DO concentrations were in equilibrium with oxygen in the atmosphere.



### 2. 2. Water masses

#### 2. 2. 1. Identification of water masses

An analysis of T$_\theta$-S diagrams was carried out for the five cruises; T$_\theta$ and S were converted to conservative temperature ($\Theta$) and absolute salinity (S$_A$) as described by McDougall and Barker (2011). For water mass identification, in this work first used the limits described by Vidal *et al*. (1994), Morrison *et al*. (1983) and Nowlin *et al*. (2001) and the recent classification proposed by Portela *et al*. (2018), as shown in figure 1a.

#### 2. 2. 2. Seasonal variation

Two of the five cruises took place during the late fall and winter (2010 and 2013), and three during summer (2011, 2015, and 2016). Since sampling in winter and summer covered approximately the same region of the GoM (Fig. 1c and 3), in this work could perform a separate seasonal analysis of hydrographic and geochemical characteristics for densities lower than 26 kg·m$_{-3}$ in the $\Theta$-S$_A$ diagrams using the Portela *et al*. (2018) classifications (Fig. 2). DO was incorporated into the diagrams to evaluate the role of seasonality on its vertical distribution in relation to water masses. It was noted that the depth of the 26 kg·m$_{-3}$ isopycnal varied by more than 100 m regardless on the time of year (Fig. 3 and Supplementary Fig. 2).

#### 2. 2. 3. T$_\theta$-S patterns above 26 kg·m$_{-3}$

Four patterns were visually identified in the T$_\theta$-S diagrams by focusing on the most distinctive characteristics for densities less than 26 kg·m$_{-3}$ (Fig. 4). The four distinct T$_\theta$-S



patterns (indicated by parallelograms) shown in Table 1 and figure 4 had the following
characteristics:

•   The blue Tθ-S pattern was characterized by a subsurface salinity maximum and lower

concentrations of DO associated with the Subtropical Underwater (SUW) (Fig. 2b

and 4).

•   The pink Tθ-S pattern was characterized by shallow fresh waters (low than 36; see

Table 1) that are likely associated with river inputs and their offshore transport. In

this study, this water mass is referred to as Freshwater Influenced Surface Water

(FISW) (Fig. 2b and 4).

•   The green Tθ-S pattern was observed during summer cruises and was characterized

by a wide range of temperatures (23.7 to 27.5ºC; see Table 1) and salinity, and a

subsurface DO maximum (H 232 µmol·kg-1) at a density of approximately 24.5 kg·m-

3. This pattern is heavily influenced by the CSW.

•   The red Tθ-S pattern was observed during winter and had a narrow salinity range

(36.4 to 36.6; see Table 1), indicating the limited influence of the CSW coupled with

seasonally lower temperatures (22.9 to 23.2 ºC; see Table 1). This so so-called Gulf

Common Water (GCW) is closer to the surface during winter.


Finally, in this work carried out a reclassification of the range limits for the water masses
lighter than 26 kg·m-3. This reclassification was done using a Matlab program that separated
and binned the data based on the four Tθ-S patterns previously described (Table 1): these
were then independently plotted to fit individual Tθ-S patterns ranges of the existing





classification established by Vidal *et al*. (1994), Morrison *et al*. (1983) and Nowlin *et al*.
(2001). A final readjustment was done based using $T_\theta$-S patterns analysis of the existing
thermohaline ranges ($\sigma_\theta$, $T_\theta$, S; Table 1) and the DO concentration of the water masses that
were observed in the $T_\theta$-S diagrams. An extended description of the code with the criteria for
classification is provided in Appendix A.

**2. 2. 4 Analysis of the vertical variability of $\sigma_\theta$, $T_\theta$ and DO in surface waters**
Sections of the vertical distribution of $\sigma_\theta$, $T_\theta$ and DO were made for each cruise (2010, 2011,
2013, 2015 and 2016, Fig. 5a-j and 6) to examine differences in the density, temperature and
DO to arising from different oceanographic conditions (Fig. 5 and 6).

**2. 4. Analysis of chemical variables**
To determine the concentration of DIC, coulometric methods were used following the
methodology described by Johnson *et al.* (1987). Reference materials were provided by the
laboratory of Dr. A. Dickson of Scripps Institution of Oceanography. The accuracy obtained
with respect to the reference material was $\pm$ 2 $\mu mol \cdot kg^{-1}$ with a precision of $\pm$ 1.5 $\mu mol \cdot kg^{-1}$.
To quantify the concentrations of combined nitrite and nitrate ($NO_2^-$ + $NO_3^-$, hereafter,
nitrate) present in the samples from the winter 2010 and 2013 cruises, a Skalar SAN Plus
autoanalyzer was used. The reference material MOOS-2 was obtained from the National
Resource Council Canada. The analytical precision was better than 5% for nitrite and nitrate
combined. For the quantification of the summer 2015 cruise, samples were analyzed with an
AA3-HR SEAL nutrient analyzer according to the GO-SHIP Repeat Hydrography Manual
(Hydes *et al*., 2010) using seawater lots CC and CD from Kanso Co. Ltd. (KANSO Technos,



Japan) as reference materials (see description in Aoyama and Hydes, 2010). Precision is
expressed as a coefficient of variation (CV) and was 0.2% for nitrate.

In order to explore possible relationships between water masses and their nitrate and DIC
content, T$\theta$-S vs. nitrate for late fall-winter of 2010 and 2013, and summer of 2015,
(respectively) were plotted and T$\theta$-S vs. DIC diagrams for late fall-winter and summer 2011,
2015 and 2016 (respectively) were also plotted. This allowed for a seasonal comparison.

**2. 5. Absolute Dynamic Topography (ADT) maps**
Absolute Dynamic Topography (ADT) maps were generated to infer the seasonal influence
of the CSW during the different cruises as Delgado *et al*. (2019) suggest. The images are
products of the AVISO + database (Archiving, Validation, and Interpretation of Satellite
Oceanographic data) available on the website https://www.aviso.altimetry.fr/en/data. The
ADT maps only considered the time in which sampling was carried out for each cruise. In
this work, present the surface dynamics based on these ADT maps, particularly from our
winter (Feb-Mar) 2013 and summer (Aug-Sep) 2015 cruises (Fig. 10b and 10e).



**2. 6. Climatological data analysis**
An analysis of the temperature and salinity data from the climatological database CARS 2009
(CSIRO Atlas of Regional Seas; http://www.marine.csiro.au/~dunn/cars2009) was
performed to contrast climatological averages between *in situ* data from winter (February)



and summer (July). Diagrams and vertical sections reflecting 50 years of monthly July and
February $T_\theta$ and S data were plotted to identify the seasonal presence or absence of CSW.

Finally, in this work developed the new reclassification of the water masses based on the
characteristic of the thermohaline and biogeochemical variables at densities lower than 26
kg·m-3 for each identified water mass.

**2. Results**
Potential temperature and salinity showed spatial and temporal variability at densities < 26
kg·m-3 during the five sampling campaigns included in this study (Fig. 1b). The four patterns
that in this work considered relevant for the designation of new thermohaline ranges for water
masses above the isopycnal of 26 kg·m-3, namely CSW, SUW, GCW, and the FISW, are
described in the following section.

**3.1. Changes in $T_\theta$ and $\sigma_\theta$ in presence or absence of CSW**
Vertical sections of seasonal changes in potential density and potential temperature occurring
in the first H 250 m (above 26 kg·m-3) of the study area are shown in figure 5. In general, the
relatively low temperatures (T H24 ºC with $\Delta T$ < 5 ºC over densities < 26 kg·m-3; Fig. 5a,
and 5b) indicate the absence of CSW in late autumn and winter and show a more mixed
column in the first 100 m. Additionally, the density was, on average H 24.5 kg·m-3 (with $\Delta$
$\sigma_\theta$ < 1 kg m-3; Fig. 5f and 5g). These characteristics are associated with the near-surface
presence of GCW. During the summer, evidence of CSW with a temperature of H 31ºC was





observed with ΔT ± 6 °C (Fig. 5h, 5i, and 5j). On the other hand, density fluctuated from $\sigma_\theta$
= 22 to 24 kg·m-3 (Fig. 5c, 5d, and 5e).

It is noticeable that during the winter of 2013, when CSW was absent, the 24 kg·m-3 isopycnal
and the 27 °C isotherm were not observed (Fig. 5b, and 5g). In contrast, water with these
characteristics was present during the summer when CSW entered the GoM through the LC
(Fig. 5c, 5d, 5e, 5h, 5i, and 5j). Therefore, the summer characteristics of density and
temperature represent the water of Caribbean origin.

**3. 2. Subsurface maximum DO and its association with GCW**
In addition to the low density/high temperature waters typical of the CSW, in this work also
noted the presence of a summer DO subsurface maximum. Figure 6 displays transects of
vertical sections of DO for the five cruises carried out during summer 2011, 2015, and 2016
a DO subsurface maximum of H 210 to 232 μmol·kg-1 is shown to exist (Fig. 6c, 6d, 6e, 7b,
and 7c). This pattern was observed consistently in the three summer cruises. The DO
maximum was located between the isopycnals of 24 kg·m-3 and 25 kg·m-3 and can be
considered a boundary between CSW and GCW. In contrast, with the absence of CSW during
late fall (November 2010), the DO subsurface maximum was no longer clearly observable.
During winter (February/March 2013), vertical mixing homogenized the DO in the first 200
m to concentrations of 200 to 220 μmol·kg-1 (Fig. 6b).
In this work found that during summer, AOU tends towards negative values (from 2 to -26.5
μmol·kg-1; see Supplementary Fig. 1c-e), above atmospheric equilibrium and supersaturated
in waters above densities of H 24 kg·m-3. In contrast, in late autumn and winter, AOU values



in the GCW were positive at the same depths and ranged from 9 to 90 μmol·kg-1 due to the
vertical transport of subsurface water (Supplementary Fig. 1a-b). This suggests that DO and
AOU profiles can be used as criteria with which to separate the CSW from the GCW.

**3. 3. Description of water masses identification using the new classification.**
To readjust the thermohaline ranges corresponding to CSW and GCW, oxygen was used as
a tracer to separate these two water masses. It is important to note that the thermohaline
ranges associated with the SUW were not modified because this water mass is only detected
inside the LC. The thermohaline and chemical characteristics of each water mass are
described in the following sections.

**3. 3. 1. Subsurface Underwater (SUW).**
Figure 7 shows the Tθ and S data above the isopycnal of 26 kg·m-3 as well as the new limits
of salinity and temperature of surface waters (see Table 2). Figure 7a, shows typical
oceanographic characteristics of water from the Caribbean, including the horseshoe structure
present in Tθ-S diagrams that describe the SUW (Fig. 2 and 4). The principal thermohaline
characteristic of the SUW is the presence of a salinity maximum (H 36.9) paired with a
relative oxygen minimum (H 137 μmol·kg-1) located between 150 and 250 m (Fig. 7a). In
this work found that SUW typically occurs in summer between 100 to 250 m and transports
low oxygen water into the GoM (Table 2).
**3. 3. 2. Caribbean Surface Water (CSW)**
CSW was only detected during the summers of 2011, 2015 and 2016. DO concentrations
varied between 180 to 190 μmol·kg-1 within the top 30 m of the water column. Surface water



above the 24 kg·m-3 isopycnal that includes the full range of thermohaline properties needs
to be better defined. The T and S ranges in this work propose for this water mass are:
temperatures between 27 and 32 ºC, salinities between 36 and 36.8, and a DO concentration
range of 180 to 220 μmol·kg-1 (Table 2). The presence of CSW can be observed from
relatively high salinities (up to H 36.8) accompanied by relatively high surface temperatures
of approx. 30 ºC (Fig. 7a, 7b, and 7d).

**3. 3. 3. Gulf Common Water (GCW)**

The surface water between the 24 and 26 kg·m-3 isopycnals also needs to be defined by
including the subsurface DO maximum as the upper limit of GCW. In this work propose new
range limits for GCW to be temperatures between 20 to 27 ºC, salinities between 36.3 to 36.6
and DO between 112 to 232 μmol·kg-1 (Fig. 7c, Table 2). Brunt-Väisälä frequency analysis
confirms the late fall data from 2010 and winters 2013 indicated vertical mixing in the first
200 m of the water column induced by season "Nortes" (not show figure).

**3. 3. 4. Freshwater Influenced Surface Water (FISW)**

The presence of the FISW was observed in summer. FISW was detected in the interior region
of the CB and was distributed along the 25 ºN transects during 2010, 2011, 2015, and 2016
campaigns (Fig. 7d). This coincided with periods of high precipitation prior to and during the
campaigns (https://smn.conagua.gob.mx/es/climatologia/temperaturas-y-lluvias/resumenes-
mensuales-de-temperaturas-y-lluvias). Based on the aforementioned thermohaline
characteristics and the distribution of this water mass, the following limits were established:
temperature between 24 to 30 ºC, salinity between 33 and 36, and DO concentrations between
180 and 220 μmol·kg-1 (Fig. 7d; Table 2).




The input of freshwater resulted in a lowering of surface salinity in the first 20 m below
approx. 36 (Fig. 7d). The temperature range was from 24 ℃ in late fall of 2010 to 30 ℃
during summer (2011, 2015, and 2016).

**3. 4. Water mass variability linked to DIC and nitrate concentrations**
In general, SUW nitrate concentrations near its $T_\theta$-S upper limit  where 0.06 μM at $\sigma_\theta$ H 24.5
kg·m$_{-3}$) increasing to H 7.1 μM a near its $T_\theta$-S bottom limit, as defined. DIC concentrations
were approx. 2098 μmol·kg$_{-1}$ and increased to H 2150 μmol·kg$_{-1}$ at H 250 m (Table 2). The
maximum nitrate concentration (H 7 μM) detected in the first 250 m was in the center core
of the SUW at $\sigma_\theta$ H 25.4 kg·m$_{-3}$ (Fig. 8a and 8c), while the DIC maximum of H 2152
μmol·kg$_{-1}$ at $\sigma_\theta$ H 25.8 kg·m$_{-3}$ coincided with DO concentrations of H 146 μmol·kg$_{-1}$ (Fig.
7a, and 8d).
As mentioned, the CSW was only detected during the summer oceanographic campaign. This
water mass was characterized by low concentrations of nitrate from 0 to 0.48 μM in the first
90 m of the water column (Fig. 8c and 9f; Table 2). Similarly, DIC in this water mass was
lower than 2090 μmol·kg$_{-1}$ (Fig. 8d; Table 2).

The GCW contained relatively high concentrations of nitrate during late fall and winter,
approx. 2 μM near 75 m. The highest concentrations of nitrate above 200 m, H 8.4 μM was
detected during this season, and it was observed within the lower limit of the GCW and the
upper limit of the TACW (Fig. 8a; Table 2). In summer, the highest nitrate concentrations of
H 1.5 μM were found near 100 m, reaching values of approx. 9.4 μM near the lower limit of



the GCW at H 210 m (Fig. 8c; Table 2). In the GCW, the vertical distribution of DIC
mimicked the nitrate profiles. During late fall and winter, DIC concentrations higher than
2080 µmol·kg-1 were found below 50 m and reached maximum values of 2172 µmol·kg-1
near the bottom depth of this water mass (Fig. 8b; Table 2).

During summer at 50 m ($\sigma_\theta$ = 24.6 kg m-3), DIC values slightly lower than 2075 µmol·kg-1
were observed to increase with depth to H 2169 µmol·kg-1 at H 210 m (Fig. 8d). The
deepening of the nutricline and carbocline observed during summer was associated with the
transport of oligotrophic waters by CSW into the GoM, with low values of nitrate < 1 µM
near the surface (Fig. 8c and 8d; Table 2).

Finally, the chemical composition of FISW depended to a large extent on the seasonality of
precipitation, fluvial inputs, and mesoscale dynamics. Stations of low salinity and low nitrate
concentrations ranging from 0.02 to 1.27 µM, and DIC ranging from 2005 to 2062 µmol·kg-
1 in the first 50 m of the water column were sampled in winter (Fig. 8a, and 8b; Table 2). In
contrast, during summer the concentrations of nitrate and DIC were slightly lower and ranged
between 0.08 to 0.34 µM, and 1968 to 2053 µmol·kg-1, respectively (Fig. 8c, and 8d; Table

2).



**4. Discussion.**
A recent detailed analysis in the central and western GoM by Portela *et al*. (2018) of water
masses from glider data, 14 cruises and Argo floats within the GoM, indicated the presence





of seven water masses. While this is an improvement, there are still some problems in the
classification and understanding of waters upwards of the 26 kg·m$_{-3}$. In this work maintain
that it is necessary to have a better understanding of how the GoM's water masses are formed
to attain a classification that gives insight into 1) the dynamics of the water masses in the
gulf, and 2) the physical mechanisms affecting biogeochemical processes, and 3) the
resulting effects within biological processes. Upwards of the 26 kg·m$_{-3}$ isopycnal,
biogeochemical variables, such as oxygen, nitrate, and DIC concentrations exhibit large
changes in concentration ($\approx$ 200 μmol·kg$_{-3}$, 0 and 9 μM, and 160 μmol·kg$_{-3}$, respectively)
that reflect the dynamic and variable characteristics of surface waters. These variations are
caused by mixing and advection, processes that are important to be identified and understood.
For this reason, it was important to reclassify the shallower water masses of the GoM by
including DO as a key tracer.
**4. 1. Reclassification of CSW and CGW using T, S and dissolved oxygen**
In this work found a noticeable presence of CSW associated with the incursion of the LC
during spring-summer as described by Delgado *et al*. (2019); this water mass was absent in
late autumn and winter. Recently, the spring-summer incursion of the LC that transports
CSW into the GoM has been confirmed, with a maximum presence in summer and a
minimum in winter (Delgado *et al*., 2019). In this work emphasize that the extended
"pulsing" by the LC and the Yucatán Current into the GoM explains the presence of CSW.
In this work attribute this absence of the CSW to the weakening of the LC.

In this work agree that the CSW increases its salt content above the 24 kg·m$_{-3}$ isopycnal from
about 36 at its entry into the GoM in the Yucatan Channel to about 36.8 due to LCE's and



coastal upwelling (Wüst, 1964; Hernández-Guerra and Joyce, 2000; Carrillo *et al*., 2016).
Also, evaporation likely contributes to the increase in salinity, caused by an increase in
surface temperature during the summer when CSW is found within the GoM (Fig. 7a, and
7b). Previous studies have reported that the increasing stratification during the summer
(mixed layer depth < 40 m) isolates the surface layer of the water column, which results in
an increase in salinity due to intense evaporation (Zavala-Hidalgo *et al*., 2014).

Recently, Portela *et al*. (2018) redefined the T-S limits of the CSW within the GoM, renaming
it a remnant of the Caribbean Surface Water (CSWr$_a$).  They indicated that the distribution
of "CSWr$_a$" is restricted to depths of 50 and 150 m. However, from the surface to 50 m they
attributed to the influence of river discharge (Fig. 1a and 9a). In this work consider that the
top 50 m should be included in an analysis that leads to the range of values used for the
classification of this water mass. By not including the full range of salinity values, the actual
volume of the CSW within the GoM would be underestimated, affect hydrography budgets
and, potentially, estimates of productivity. Additionally, in the classification proposed by
Portela *et al*. (2018) the overlap in the thermohaline ranges of the CSW and GCW was
overlooked (see figure 2 of Portela *et al*., 2018).

In this work, solved the overlap problem based on the fact that the CSW is closely linked to
the LC by the Yucatán Current input to the CB. In this work suggest that the overlap in the
characteristics of the CSW and GCW that was not addressed by the Portela *et al*. (2018)
classification can be addressed by considering the subsurface DO maximum. Our analysis
revealed the existence of a subsurface DO maximum, which allowed us to separate the upper
limit of the GCW from the bottom of the CSW. However, in this work suggest that the





mechanism by which do behaves conservatively is as follows: during autumn-winter when
the incursion of LC is minimal, the GCW is distributed at the surface. Intense winds are
known as "Nortes" occur during this period and intense mixing takes place in surface waters
of the GCW, resulting in the homogenization of all properties. The oxygen concentration
measured during the winter months was approx. 220 µmol·kg$_{-1}$ (Fig. 6a-b and 7c). During
spring-summer, the LC advects the warm, oligotrophic waters of the CSW into the interior
of the GoM on top of the GCW. This water has a lower DO concentration than that found in
the surface waters of GCW in winter, which is caused by temperature-related differences in
solubility (Benson and Krause, 1984). The warm water of the CSW induces stratification that
limits the exchange of oxygen with the underlying GCW (Fig. 6a-c, 7a, and 7c). The
boundary between both water masses is therefore indicated by the maximum subsurface DO
concentration (Fig. 6). In this work estimate that the DO concentration difference is approx.
50 µmol·kg$_{-1}$ (180 to 230 µmol·kg$_{-1}$ see figure 6), and can this difference can be explained by
differences in solubility, ruling out that the DO maximum is associated with photosynthesis.
This is supported by a depth difference between the peak of maximum fluorescence and the
maximum subsurface DO, maximum fluorescence occurs below of the subsurface DO
maximum (Supplementary Fig. 3). Also, during summer cruises, the AOU in the CSW tends
towards negative values (Supplementary Fig. 1c-e), these are usually found above densities
of approx. 24 kg·m$_{-3}$ (Fig. 5), from a greater exchange with the atmosphere. In contrast,
during late autumn and winter, the AOU presented positive values due to more respiration
within the GCW (Supplementary Fig. 1a-b).

**4. 1. 2. On the formation of GCW**



The surface presence of the GCW in the autumn and winter is caused by: 1) the absence of
CSW due to the retraction of the LC, and 2) the strong winds that result in a well-defined and
deep (100 m) mixed layer. This last observation was previously pointed out by Nowlin and
MCLellan (1967), Elliott (1979,1982), Vidal *et al.* (1994), and Portela *et al.* (2018). It has
been suggested that the formation of the GCW originates from the erosion of the SUW (Vidal
*et al*., 1992, 1994; Portela *et al*., 2018). However, our results suggest GCW formation
originates from the mixture of the remains of CSW and SUW within the GoM when the LC
is retracted. During fall and winter, the remnant of these water masses in the interior of the
gulf is mixed with TACW to form GCW.

During winter, when the CSW is absent, the TACW was also shallower than in summer. The
proximity of the TACW to the GCW facilitates the vertical exchange of chemical properties
towards the surface. Convective mixing leads to low DO concentrations of the TACW to be
reflected in the GCW, as well as causing an observable increase in nitrate and DIC
concentrations (Fig. 6). Furthermore, observations by satellite of the GoM found maximum
concentrations of chlorophyll in winter (Pasqueron *et al.*, 2017). This is in agreement with
Damien *et al*. (2018), who found a winter chlorophyll concentration increase explained by
the amount of nutrient injected into the euphotic layer by the dynamic of the winter mixed-
layer.

**4. 1. 3. Freshwater Influenced Surface Water (FISW)**
The presence of FISW reported in this study during the summers in the central region (24º-
25ºN, 95.6º-88ºW) is likely due to river inflows, precipitation and offshore transport. In the
central region of the GoM, relatively low salinities were measured that can only be explained



by the contribution of freshwater from rivers or precipitation. For example, in the central
stations located along 25 ºN, salinities of approximately 33.1 were detected in the first 20 m
of the water column, which would lead to the formation of FISW (Fig. 1c). These freshwater
inputs were also reported by Portela *et al.* (2018), who detected the influence of low salinity
waters (33 g·kg$_{-1}$) within the first 50 m in the central gulf. These low salinities have been
attributed to the influence of freshwater inflow from rivers to the continental shelf in the
northern GoM and transport to the central gulf by anticyclonic eddies; hence, low surface
salinities can be found hundreds of kilometers from the river source (Morey *et al.*, 2003a;
Morey *et al.*, 2003b; Jochens & DiMarco, 2008, Brokaw *et al.*, 2019).

In the northern GoM, the Mississippi and Atchafalaya rivers flow into the GoM. Their
outflow is generally transported westward along the Louisiana shelf during the summer
months (Cochrane and Kelly, 1986; Ohlmann and Niiler, 2005; Smith and Jacobs, 2005) in
response to predominant winds from the north and east (Wang *et al.*, 1998). Besides, it has
been reported that these rivers have their highest inflow during the spring/summer (Morey *et*
*al.*, 2003a).
In the southern GoM the Tonalá, Coatzacoalcos, and Usumacinta rivers flow into the region
bordering CB. It has been reported that the propagation of low salinity filaments can be
caused by local circulation resulting in a salinity gradient from coast to ocean (Vidal *et al.*,
1994). In this work also observed the FISW as part of a salinity gradient of 35.4 to 36.3 that
extended from the edge of the shelf toward the ocean, particularly during the winter. Also, a
decrease in offshore salinity was attributed in the coastal region of the CB to freshwater input
by Vidal *et al.* (1994); FISW was also detected at stations closer to the coastal region of the
CB in the three summers oceanographic camping. It may also be noted that this type of water



was observed in the semi-permanent cyclonic eddy reported by Nowlin (1972) and Pérez-
Brunius *et al*. (2013), which could contribute to the transport of the FISW in the Campeche
region during both summer and winter.

Concerning the biogeochemical role of the FISW in the surface waters within the GoM, the
following questions remain: 1) what is its influence of the FISW in the first 20 m? and 2)
what is its influence in the central GoM?

These questions highlight the need to carry out studies of biogeochemical processes at
smaller scales to determine their role within the GoM. Undoubtedly, it is also important to
carry out studies at the river mouths to determine the flow of nutrients and organic matter to
the gulf.



**4. 1. 4. The surface water masses modulate the depth of the nutricline**
One of the biological implications of the presence of CSW is that it is oligotrophic reaching
down to 90 m in spite. This can be seen in figure 10, wherein this work compares the vertical
distribution of nitrate and density with ADT maps for summer 2015 (when mesoscale eddies
were abundant) and winter 2013 (when the number and spatial extent of eddies were smaller).
During summer, a near-surface incursion of low-density water associated with the CSW was
observed (white line Fig. 10a). This incursion brought water with oligotrophic characteristics
to depths shallower than 70 m (nitrate from 0 to 0.48 µM; DIC H 1978 µmol·kg-1, Fig. 8).
Nitrogen fixation process uses to be present on this oligotrophic surface North Atlantic Ocean



waters (Montoya *et al*., 2002). The horizontal distribution of the concentration of nitrate and
DIC was reduced by stratification following the entrance of the LC that transport the CSW
into the interior of the gulf. In winter, the absence of the CSW is accompanied by a well-
mixed density distribution in the first 200 m as the GCW predominates (Fig. 10d). Higher
nitrate (0.02 to 13.7 μM) and carbon (> 2036 μmol·kg$_{-1}$) concentrations were observed near
the surface above depths 75 m.

Therefore, the alternating absence or presence of the CSW is related to the nutricline depth;
in summer when CSW overlies the GCW, the nutricline is deepest (Fig. 10). In winter, when
the GCW predominates and the TACW is shallower, deeper and well-defined, the nutricline
is found closer to the surface. The importance of this redefinition of the water masses
contributes to a better understanding of their role in the dynamics of nutrients (and carbon).
Finally, an analysis was carried out using the CARS2009 database (CSIRO Atlas of Regional
Seas) in order to evaluate the temporal changes of the CSW and the GCW. Figure 11 contrasts
climatological averages between winter (February) and summer (July). The T$_{\theta}$-S diagram, as
well as the vertical sections, show that CSW is only evident during the summer while during
the winter only the GCW is detected from the surface to approximately 200 m deep. This
supports our suggestion that the seasonal extension and retraction of the LC favors the
formation of the subsurface maximum of DO during the summer and disappears in winter.
Figure 11 shows that during the presence of the CSW cause a deepening of the nutricline
during the summer to H 150 m in contrast to winter when the nutricline rises toH100 m.

The analysis of the CARS2009 climatological data confirms the importance of CSW in
affecting the near-surface biogeochemical characteristics of the GoM. Both the cruise data



and the CARS2009 climatological data sets affirm that the DO subsurface maximum can be
used to define the upper limit of the GCW. During the summer months, with the entry of LC
and dissipation by eddies, the presence of CSW dominates in the first 100 m, potentially
having an impact on the primary productivity of the GoM, as indicated throughout this work
and by other authors (Nowlin & McLellan, 1967; Tanahara, 2004; Schmitz, 2005; Delgado
*et al.*, 2019),

**5. Conclusions.**
A re-classification of the water masses above the 26 kg·m-3 isopycnal was carried out
resulting in a modification of the present thermohaline ranges defining the CSW and GCW
water masses. For the re-classification of the CSW and the GCW, DO concentrations were a
key indicator of water mass limits. In addition, another water mass, the FISW, formed by the
influence of the freshwater inputs, was included in the new classification.
CSW was detected only during the summer with a vertical spatial domain encompassing the
first 90 m and featured warm waters, high salinities, non-detectable nitrate concentration, and
negative values of the AOU. It was also found that the lower limit of this water mass is
delimited by a maximum subsurface DO. The presence of this subsurface maximum was
found only in the summer and separates the CSW from the GCW. Likewise, the presence and
absence of CSW was found to modulate the depth of the nutricline and likely influences
primary productivity.

In winter, the replacement of the CSW by the GCW affected the biogeochemical composition
of surface water, specifically with an increase in nitrate concentrations, positive values of
AOU and a decrease in surface temperatures. The TACW lies below the GCW and is closer



to the surface than during the summer, contributing to nutrient availability and low DO near
the surface.

The SUW was detected during most of the year only in the vicinity of the Yucatán Channel
and along the region of influence of the LC. This mass of water stands out for its subsurface
salinity maximum, low DO and high nitrate and DIC concentrations when compared to CSW.

Finally, in this work proposed new criteria for the identification of the near-surface FISW.
This was detected in the central oceanic region of the GoM indicating the contribution of
precipitation and offshore transport of river discharge waters from the northern GoM
(Mississippi and Atchafalaya).

**Data availability**
The data is not available at the moment.
**Author contribution**
The study was conceived by all co-authors. GYCD carried out the sampling on board *R/V*
*Justo Sierra* cruise XIXIMI-1 to XIXIMI-5 and the analytical work in the laboratories at the
Oceanological Research Institute (IIO) México. This work proposes a new reclassification of
the surfaces water masses in the GoM for the long-term effects on conditions biogeochemical
processes. GYCD prepared the manuscript with substantial contributions from all co-authors.
**Competing interest**



The authors declare that they have no conflict of interest.
**6. Acknowledgements**
This study is a contribution of the Consorcio de Investigación del Golfo de México (CIGoM)
through the project 201441 "Implementación de redes de observación oceanográficas
(físicas, geoquímicas, ecológicas) para la generación de escenarios ante posibles
contingencias relacionadas a la exploración y producción de hidrocarburos en aguas
profundas del Golfo de México" funded by Secretary of Energy (SENER)-National Council
of Science and Technology of Mexico (CONACyT) Hydrocarbons Fund. Altimeter products
were produced by Data Unification and Altimeter Combination System available on the
AVISO (Archiving, Validation and Interpretation of Satellite Oceanographic data)
https://www.aviso.altimetry.fr/en/data. Wind Stress, Geostrophic and Ekman Currents were
extracted from GEKCO (Geostrophic Ekman Current Observatory, Sudre et al., 2013)
http://www.legos.obs-mip.fr/members/sudre/gekco_form with support from LEGOS. In
particular for wind stress GEKCO product, they were used these three sources for 01/01/1993
- 27/10/1999 period https://www.ncdc.noaa.gov/data-access/marineocean-data/blended-
global/blended-sea-winds, for 28/10/1998 - 20/03/2007 period (MWF L3 daily QuikSCAT
product) http://cersat.ifremer.fr and for the 21/03/2007 - 31/12/2017 period (MWF L3 daily
ASCAT product) http://cersat.ifremer.fr/data/products/catalogue. Finally, the general
features of the Gulf of Mexico Loop Current eddies were taken from
https://www.horizonmarine.com/loop-current-eddies. Also, we especially thank Dra. Esther
Portela her positive criticisms and suggestions.



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





**Appendix A: Description of the code that was developed in Matlab.**
The code developed in Matlab (ver. R2014a) uses a step scheme in which the complete data
set is initially included, which will automatically exclude with specific criteria (Table 1) the
four structures identified as follows.
This program was developed in the following manner:
**1.-** Specific criteria were assigned to the thermohaline variables ($\sigma_\theta$, $T_\theta$, S) and to the DO
variable to facilitate the identification and separation of profiles (Table 1).
**2.-** The program identified hydrographic profiles with similar thermohaline characteristics
(previously specified) and grouped them into four data subsets. Each data subset
corresponded to one of the previously identified structures. As was observed, the structures
associated with SUW (1st pattern; blue: maximum subsurface S; Fig. 4) and FISW (2nd
pattern; pink: low surface S; Fig. 4) presented extreme thermohaline characteristics, making
them easy to group. However, the structures associated with CSW (3rd pattern; green:
maximum surfaces S and T; Fig. 4) and GCW (4th pattern; red: narrow surfaces S and low T;
Fig. 4) were more difficult to group, but the DO variable proved to be the key to identification
and separation.
**3.-** To separate the patterns associated with CSW (3rd pattern) and GCW (4th pattern), two
additional conditions within the program were set and are as follows:

**3.1.** To separate CSW from GCW, DO data located between the 23.75 and 24.75
kg·m-3 isopycnal were averaged (1st condition), associated with the red profile (fourth
pattern).



**3.2.** The code calculated the average value of the DO data less than the 23.5 kg·m-3
isopycnal (2nd condition), associated with the green profile (third pattern).
**3.3.** If there are no data less than 23.5 kg·m-3 density, the code will search for the first
5 surface data to average value the DO and carry out the next step (2nd condition).
**3.4.** Finally, the program carried out a comparison between the average values
obtained from both conditions. This comparison was used to determine if the DO
values from the 1st condition were greater than those from the 2nd condition. If this
was true, the code classified the data set with the green profile characteristics (3rd
pattern). If it was not the case, the code classified it with the red profile characteristics
(4th pattern).
**3.5.** After we obtained the four structures separately, they were plotted separately and
associated with the water masses present upper the 400 m of the water column.
Subsequently, the limits thermohaline properties, DO concentrations, DIC, and
nitrates for each identified mass of water were identified.










**Table 1.** Thermohaline characteristics and oxygen values used to separate the four identified
structures (1st blue pattern; 2nd pink pattern; 3rd green pattern, and 4th red pattern) that were
used in the program developed in Matlab (ver. 2014Ra).

| $T_\theta$-S patterns | $\sigma_\theta$ (kg m$_{-3}$) | Salinity (S) | Temperature (ºC) | Oxygen (μmol kg$_{-1}$) |
|---|---|---|---|---|
| Maximum subsurface S (1st) | 25.4 - 26 | S ≥ 36.68 | 19 - 22 | 140 - 160 |
| Low surface S (2nd) | 21 - 24 | ≤ 36.0 | 24 - 31 | O$_2$ ≥ 193 |
| Maximum surface S and T (3st) | 24.8 - 25.25 | 36.4 - 36.6 | 22.9 - 23.2 | O$_2$ ≥ 185 |
| Maximum surface S and low T (4th) | 23.7 - 24.7 | 36.3 - 36.67 | 23.7 - 27.5 | 190 - 204 |


**Table 2.** General characteristics of the new classification of the surface water masses
identified based on thermohaline variables (Potential temperature [ºC] and Salinity [psu]),
DO [μmol·kg$_{-1}$], and AOU [μmol·kg$_{-1}$]. Also, the variability ranges for DIC [μmol·kg$_{-1}$],
nitrates [μM], and depths as a function of each water mass identified in the deep region of
the GoM were included.

| Water masses | ID | Temperature θ | Temperature Θ | Salinity psu | Salinity gk·g$_{-1}$ | Oxygen (μmol·kg$_{-1}$) | DIC (μmol·kg$_{-1}$) | Nitrates (μM) | Depth (m) | AOU (μmol·kg$_{-1}$) |
|---|---|---|---|---|---|---|---|---|---|---|
| Caribean Surface Water | CSW | 27 - 32 | 27.1 – 32.1 | 36.0 - 36.8 | 36.18 - 36.98 | 180 - 220 | 1978- 2090 | 0 – 0.50 | < 90 | -27 to 2 |
| Subtropical Underwater | SUW | 19 - 26 | 19.1 – 26.1 | 36.6 - 37.0 | 36.78 – 37.18 | 136 - 180 | 2098 - 2156 | 0.06 – 7.10 | 100-250 | 50 to 95 |
| Gulf Common Water | GCW | 20 - 27 | 20.1 – 27.1 | 36.3 - 36.6 | 36.48 – 36.78 | 112 - 232 | 2036 - 2172 | 0.02 – 9.40 | 0-200 Winter 50-200 Summer | 0 to 90 |
| Freshwater Influenced Surface Water | FISW | 24 - 31 | 24.1 – 31.1 | ≤ 36 | 33.28 – 36.18 | 180 - 220 | | | ≤ 20 | |












**FIGURE CAPTIONS:**

**Figure 1:** (a) Distribution of the water masses using the classification system proposed by Portela et al. (2018) using conservative temperature ($\Theta$) vs absolute salinity [$S_A$ g·kg$_{-1}$], water masses as: Caribbean Surface Water remnant (CSWr$_a$), North Atlantic Subtropical Underwater (NASUW), Gulf Common Water (GCW), Tropical Atlantic Central Water (TACW), TACWn$_a$ (nucleus), Antarctic Intermediate Water (AAIW) and North Atlantic Deep Water (NADW). (b) $\Theta$-$S_A$ vs dissolved oxygen [DO, µmol·kg$_{-1}$] diagram showing upwards of the isopycnal of the 26 kg·m$_{-3}$ using the Portela et al. (2018) classification. The data from the five cruises from 2010 to 2016 were used to generate the $\Theta$-S diagrams. (c) The coverage area for the stations analyzed (transect delimited in black lines) in the GoM from 2010 to 2016.

**Figure 2:** Seasonal comparison (late fall-winter (a) and summer (b)) of the $\Theta$-$S_A$ vs. DO [µmol·kg$_{-1}$] diagrams showing upper waters ($< 26$ kg·m$_{-3}$) using classification Portela et al. (2018) considered: CSWr$_a$, NASUW, GCW, and TACW. To generate $\Theta$-$S_A$ vs. DO [µmol·kg$_{-1}$] diagrams in this work used data from the five cruises where the years 2010-2013 (late fall and winter) were separated from the years 2011, 2105 and 2016 (summer).

**Figure 3:** A comparison of winter (a) and summer (b) conditions of the variability of the depth of 26 kg·m$_{-3}$ density field in the GoM (in situ hydrographic data collected in February/March 2013 and August/September 2015, respectively)

**Figure 4:** The $\Theta$-$S_A$ diagram shows the four characteristic patterns like the average considering the five cruises (blue: maximum subsurface S; pink: low surface S; green:



maximum surfaces S and T; and red: maximum surfaces S and low T) identified for the five
cruises using the ranges shown in Table 1. The Portela et al. (2018), classification was
incorporated into the $\Theta$-$S_A$ diagram to determine if the patterns identified to fit the above
classification (water masses: CSWra, NASUW, GCW, and TACW).

**Figure 5:** The vertical distribution [250 m] of potential density [kg·m-3] and potential
temperature [ºC] are shown for the late fall of 2010 (a and f), winter of 2013 (b and g) and
summers of 2011 (c and h), 2015 (d and i), and 2016 (e and j). The location of the transect is
shown in figure 1c.

**Figure 6:** The vertical distribution [250 m] of dissolved oxygen [µmol·kg-1] are shown for
the late fall of 2010 (a), winter of 2103 (b) and summers of 2011 (c), 2015 (d), and 2016 (e).
The white contours indicate the lower limit of CSW [24 kg·m-3] and GCW [26 kg·m-3] in all
sections. The location of the transect is shown in figure 1c.

**Figure 7:** This figure shows the new classification of the water masses with the adjustments
to the thermohaline range limits based on the distribution that the four patterns in Figure 2c.
(a) The Tθ-S vs DO [µmol·kg-1] diagram shows the profiles with SUW characteristics. (b)
Tθ-S vs DO [µmol·kg-1] diagram presents characteristics particular with SCW (c) Tθ-S vs DO
[µmol·kg-1] diagram associated with GCW. (d) Tθ-S vs DO [µmol·kg-1] diagram associated
to the water mass called Freshwater Influenced Surface Water (FISW).

**Figure 8:** (a) Tθ-S vs nitrate [NO2- + NO3-, µM] and (b) Tθ-S vs dissolved inorganic carbon
[DIC, µmol·kg-1] diagrams corresponding to the winters of 2010 and 2013. (c) Tθ- S vs nitrate





[μM] and (d) Tθ-S vs DIC [μmol·kg-1] corresponding to summer 2015.

**Figure 9:** Comparison of the classification system proposed by (a) Portela et al. (2018) and
(b) this study. Shows the Θ-S$_A$ vs. DO [μmol·kg-1] diagram showing upwards of the isopycnal
of the 26 kg·m-3 using the reclassification proposed in this work. The names of the water
masses used in this work are: Caribbean Surface Water (CSW), Subtropical Underwater
(SUW), Gulf Common Water (GCW), and the Freshwater Influenced Surface Water (FISW).

**Figure 10:** The vertical distribution [250 m] of potential density [kg·m-3] is shown for the
summer of 2015 (a) and winter of 2013 (d). The white contours indicate the lower limit of
CSW [24 kg·m-3; (a)] and of GCW [26 kg·m-3; (d)] in both sections. The ADT maps show
the trajectory of the summer (b) and winter (e) sections of each cruise. The nitrate profiles
[μM] (c=summer; f=winter) only include the stations that are found within the trajectory
traced in the ADT maps for each cruise. The blue color points indicate the stations that are
found outside of the areas influenced by anticyclonic rings while the red color points denote
stations located in the area of influence of the anticyclonic gyres.

**Figure 11:** Θ-S$_A$ vs. DO [μmol·kg-1] annual diagrams from February (a) and July (b) showing
the reclassification proposed in this work. Data derived from the CARS-2009 database.
Annual vertical sections (-95.5 to -86.5 ºW, 25 ºN; the section shown in figure 1c from the
station C to D) of oxygen [μmol·kg-1] concentration and nitrate [μM] for February (c, and d),
and July (e, and f) derived from the CARS-2009 database




**FIGURE 1:**

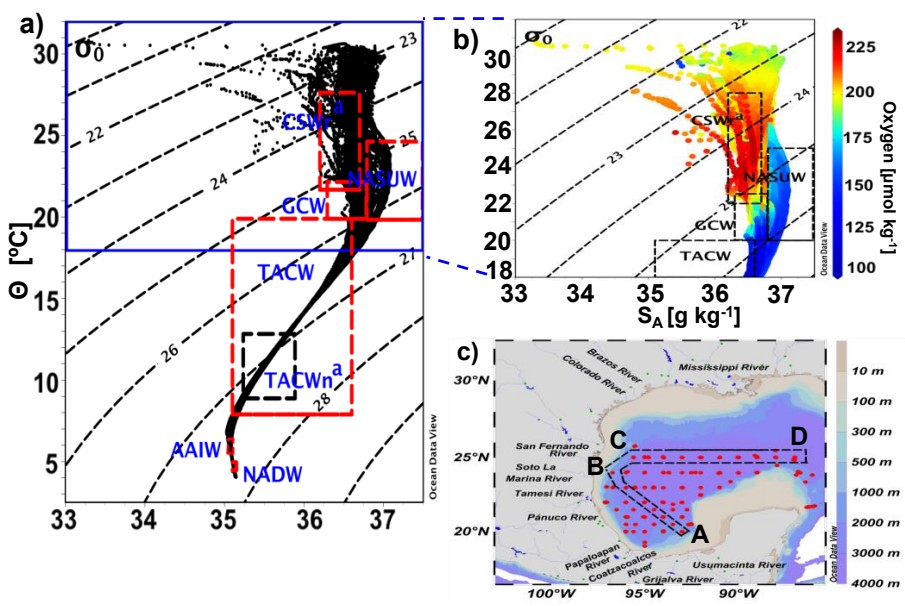


**FIGURE 2:**

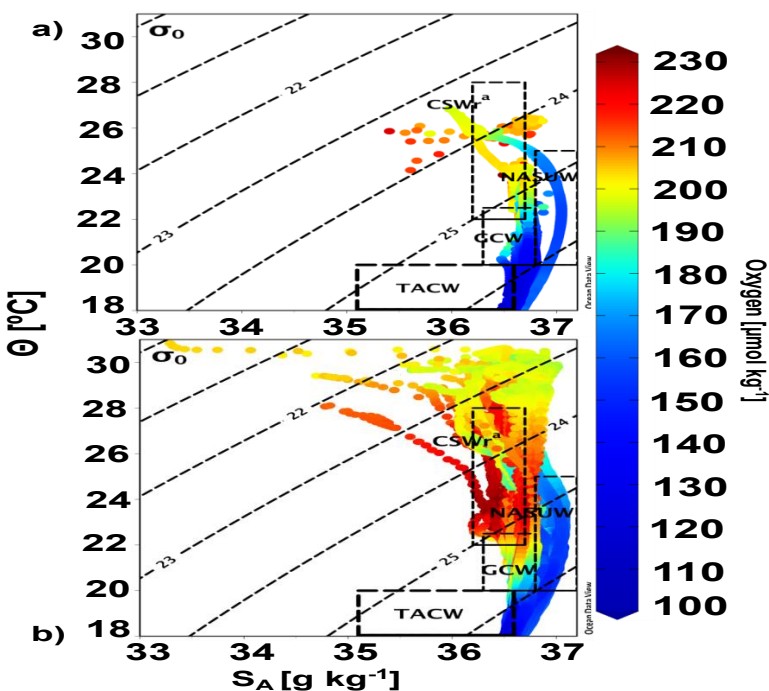




**FIGURE 3:**

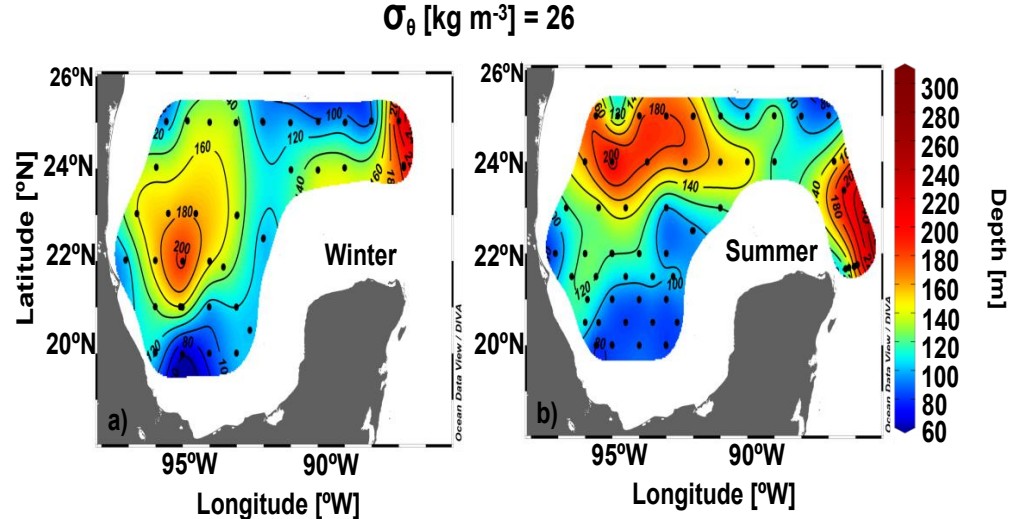



**FIGURE 4:**

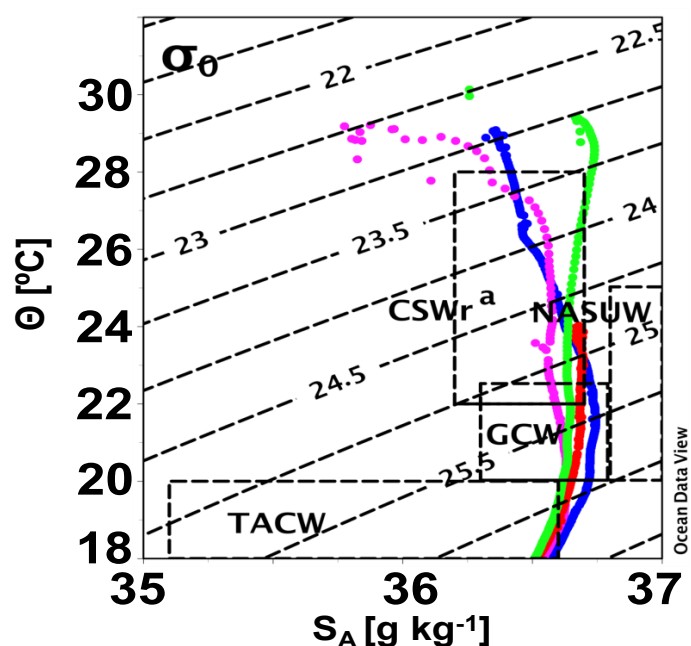




**FIGURE 5:**








**FIGURE 6:**

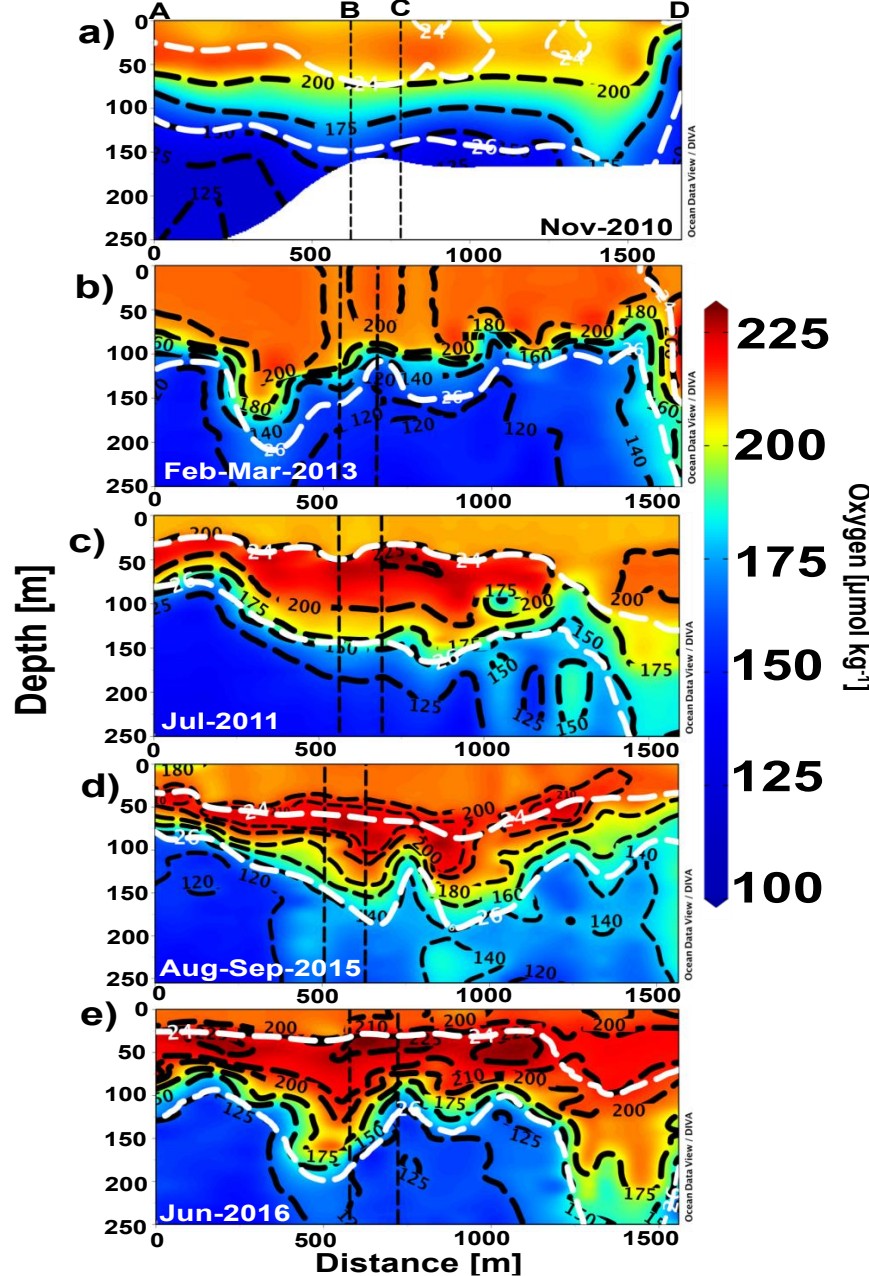





**FIGURE 7:**

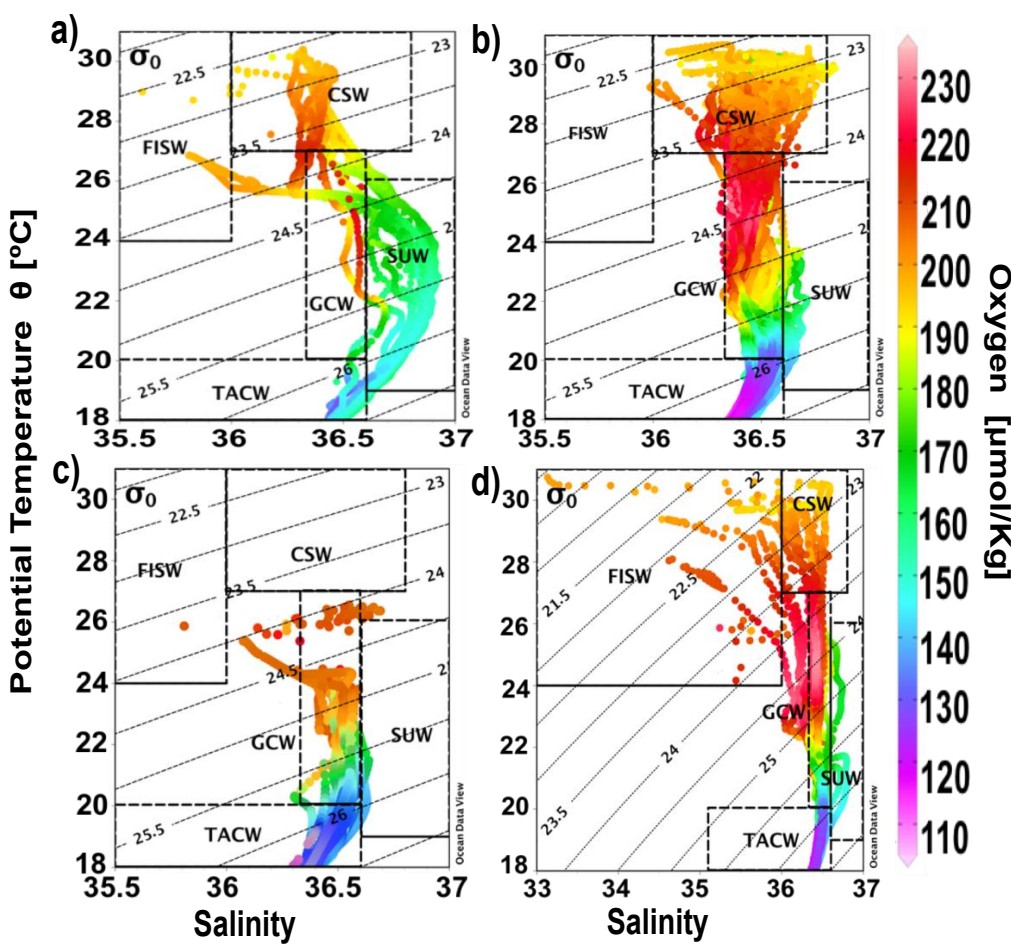












**FIGURE 8:**

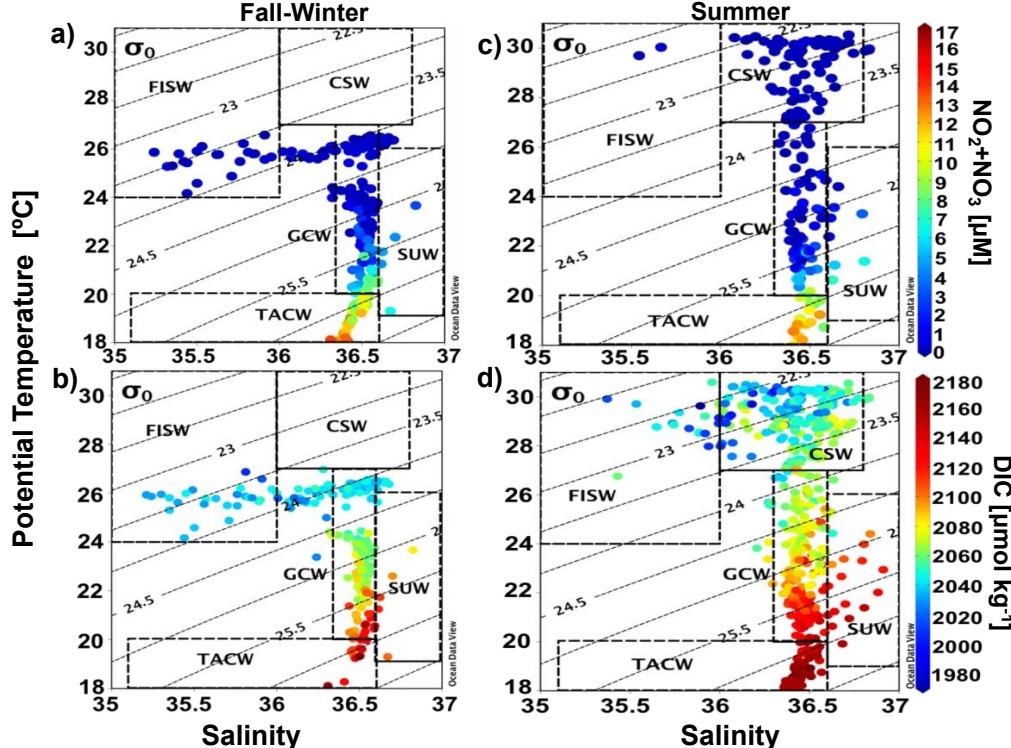














**FIGURE 9:**

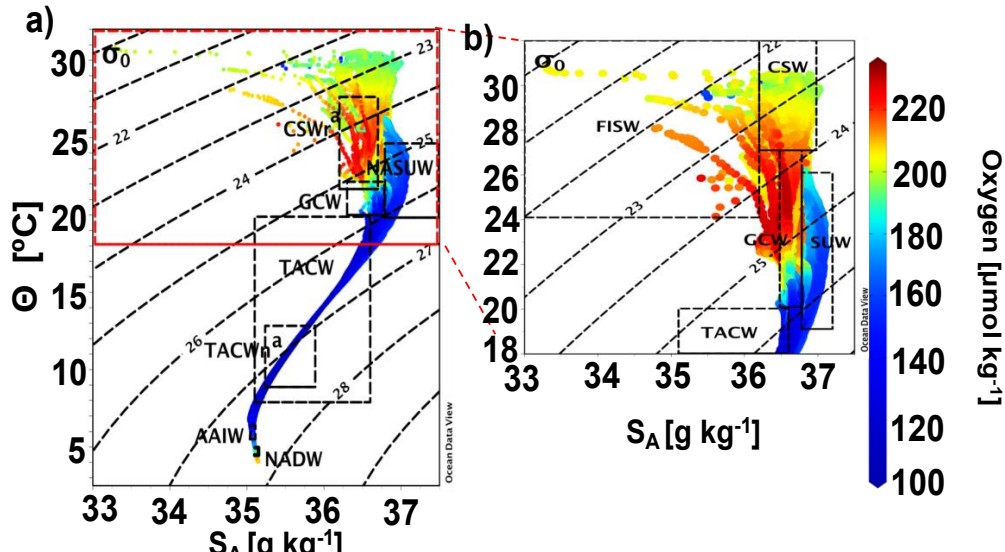

**FIGURE 10:**

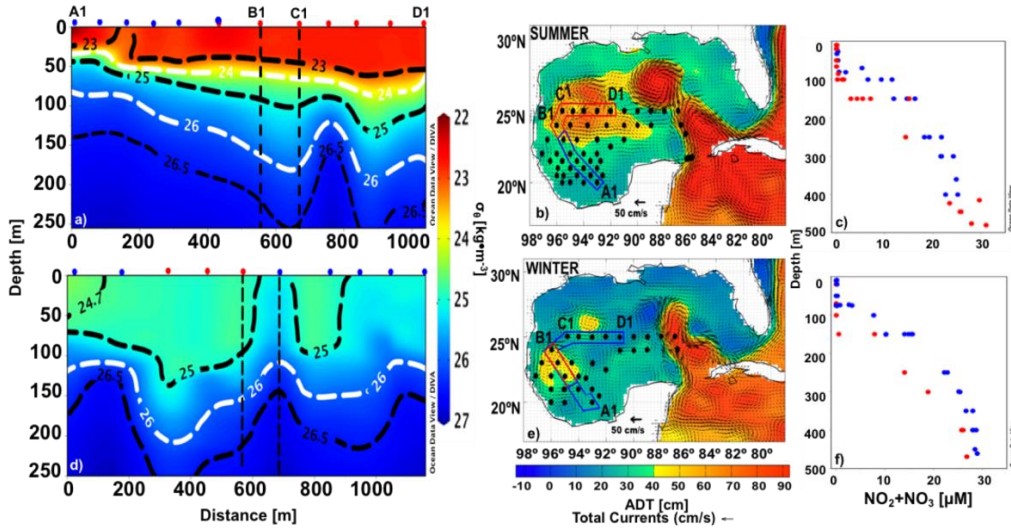



**FIGURE 11:**

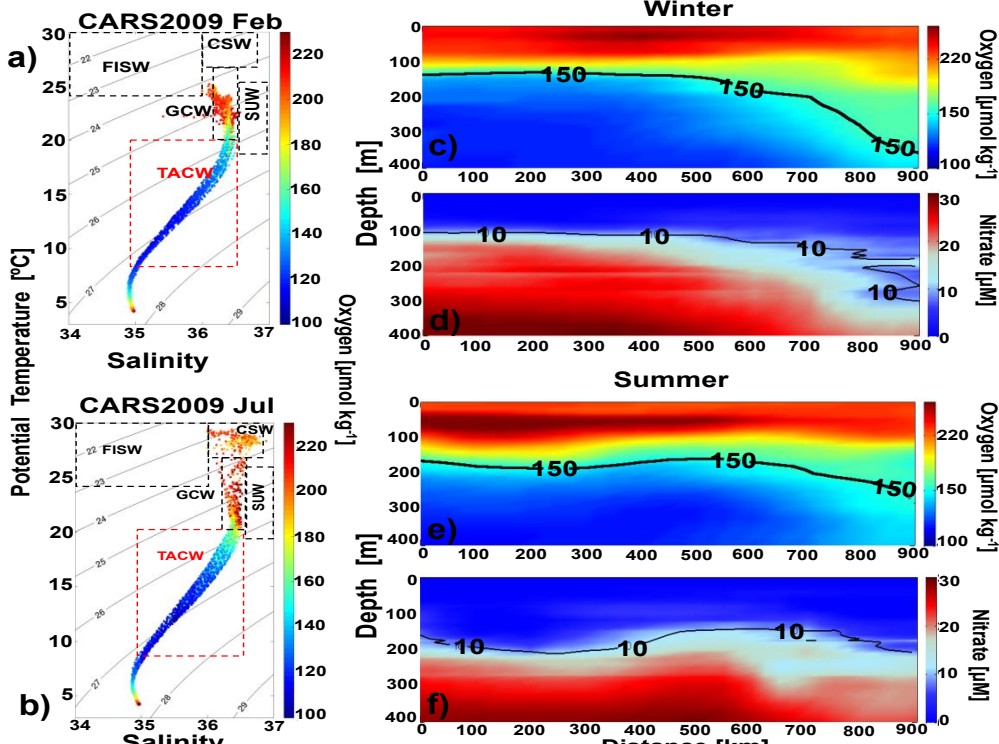



