# Peer review of "A New Characterization of the Upper Waters of the central Gulf of México"

_Biogeosciences, 2019_

## Referee Comment (RC1) · Anonymous Referee #1 · 12 Oct 2019

General Comment Based on hydrographic and biogeochemical measurement obtained during five oceanographic cruises, this article proposes a redefinition of water masses over the upper ocean in the central Gulf of Mexico. A new water mass for the most surface layer that considers the effects of fresh water from precipitation and river runoffs is introduced. The data sets used in this research are interesting, as they captured the seasonal variability in hydrographic and biogeochemical properties over the Loop Current (LC), anticyclonic mesoscale eddies that separated from the LC, and cyclonic mesoscale circulations such as the Bay of Campeche Cyclone, and some frontal LC

cyclones. The attempt to eliminate overlaps between some of the water masses, and the introduction of the new surface water mass, is appreciated. However, the justification on the need to change the conventional ranges for the gulf's water masses is not convincing, and the way the water masses were defined is weak. Below, I am including a number of scientific issues that need to be addressed for making a more convincing case, and for improving the presentation and readability of the manuscript.

Specific Comments 1. The definition of water mass FISW does not satisfy conventional approaches to name the body of water and delineate its properties. a) Note that a water mass is defined as a body of water with a common formation history. Given that 'FISW' is under the continuous action of highly dynamic and variable forcing (wind stress, insolation, air-sea fluxes, precipitation, and mesoscale eddy dynamics), is it possible to attribute a common formation history to this body of water? b) Since the name of the water mass usually relates to its major area of residence, the name FISW is not appropriate for this potentially new water mass. c) A water mass is often found in regions well beyond its formation region. Is this condition satisfied in the case of FISW? I do not think so. This condition is difficult to be verified. d) A water mass can be identified away from its formation region because its elements retain its properties, in particular its potential temperature and salinity. Given that FISW extends over surfaces waters where irreversible vertical mixing is continuously changing the temperature of the body of water (diabatic turbulent process), temperature is not a conservative property over these surface body of water. Note that classical water masses that form in the surface usually sink into deeper waters away from surface forcing, which allows them to retain its original properties for long periods of time over grate distances. e) In order to accurately define a water mass, it is necessary to include information about its standard deviation; some water masses only require a single combination of T-S and its standard deviation, while delineating other water masses may require defining a T-S relationship and an envelope for the standard deviation. This requirement must be satisfied in the definition of FISW (no information is given about its standard deviation). Is its standard deviation small enough?

2. The definitions of water masses presented here need to be compared against historic definitions that are well established in the scientific community. Modifying table 2, and creating a new figure showing the different ranges of the water masses reported in the literature, will help in evaluating whether we need a new definition of the Gulf's water masses.

3. Why is the overlap between CSW and GCW a big deal? There is always a transition region between water masses in every ocean (that is why we need a standard deviation in trying to isolate the dominant characteristics of the water mass). In order to justify the idea that we need to get rid of this overlap, this manuscript needs to quantify the error related to the conventional and new definitions, including a quantification of the effects on the density field. Are these errors significant, or are both at the noise level? If errors from both definitions are at the noise level, no new definition is needed. The manuscript is missing an in-depth review of the-state-of-the-art on the formation of GCW.

4. It is not clear how the LC cycle can be used to eliminate the overlap between CSW and GCW since the LC does not transport GCW. This is an important issue in the approach presented here. Also note that Caribbean anticyclones can also make it into the Gulf transporting CSW. Moreover, atmospheric forcing could erase the CSW signature in winter over the Gulf. Some misleading statements regarding this issue are: a) This explanations in 455-459 and 553-554 are convoluted. Since the LC does not transport GCW (water mass originated in the GoM), and the CSW is presumably only found in the LC, how the CSW ends up on top of GCW? Something does not make sense here. b) Too much emphasize is put on the idea that the "weakening" of the LC is associated with the absence of CSW in the GoM(423). However, previous studies (cited in the present article) claim that the absence of CSW is becasue this water is continuously transformed in the GoM by wind forcing. Since the latter idea weakens the hypothesis that the LC cycle can be used in dealing with the overlap between CSW and GCW, this issue needs to be addressed in detail. Is CSW absent

in winter in both the GoM and Caribbean Sea? If it is missing also over the Caribbean Sea, then atmospheric processes control the variability of this water mass over both the Gul and the area of formation. c) In lines 428-430 it is claimed that there is a salinity contribution to CSW in the GoM. Is not supposed that CSW acquires its distinctive high salinity values over the Caribbean Sea? A local addition of salinity within the GoM is against the conventional definition of water mass.

5. I understand that water masses formed at the surface at higher latitudes retain its DO because they sink and move away from regions of intense atmospheric forcing. However, in the case of water masses that remain in the surface, is it valid to use DO for characterizing their properties? I am not sure about this, since intense vertical mixing acting over these bodies of water makes them diabatic (their properties are non-conservative). Note that DO is a function of temperature, and temperature is non-conservative in water parcels over the ocean mixed layer and upper thermocline. Also note that the LC cycle is not needed to have the variability in DO documented here (562-564). It needs to be shown the variability in DO is not caused by atmospheric forcing in surface water masses; otherwise, it cannot be used in characterizing surface water masses.

6. An important analysis and methodology for redefining the water mases are given in Fig. 4 and appendix A. These approaches can be significantly simplified by satisfying the conditions listed in item 1 above; using the standard deviation can be particularly helpful.

7. What are the source of nitrite and DIC contained in FISW? Is the seasonal variability in these properties related to vertical mixing (and cooling of the sea surface), since these two chemicals depend on temperature? Because these properties reflect the dynamic and variable characteristics of surface waters (409-412), can they be used in delineating water masses? They are clearly impacted by the seasonal cycle of insolation and vertical mixing over the upper ocean, and likely also by local biogeochemical processes.

8. Where is the analysis of the Brunt-Vaisala frequency (345-247) being shown? Rather than buoyancy alone, it is the criticality of the Richardson number (Ri<1/4) that is used to identify periods of vertical mixing. In addition to the buoyancy frequency, measurements of horizontal current vertical shear are also needed in the computation of Ri.

9. Another possibility for explaining the seasonal change in the nutricline and carbocline (387-390), is that these properties are a function of the seasonal cycle of the wind stress and insolation since these properties clearly are a function of temperature as per Fig. 8.

10. The vertical exchange of chemical properties between water masses discussed in 484-486 can occur by diffusion (very low time scale), or by diapycnal mixing that requires vertical mixing and water mass transformation. What is the more likely mechanism for explaining this conundrum? Again, the introduction is needs an in-depth discussion on the formation of GCW.

11. Note that the Mississippi River plume (508-510) also extends southward into the LC and associated eddy field; this plume can also leave the GoM through the Florida Straits. This topic needs a review of the state-of-the-art, since river runoff can be an important contribution to FISW.

Technical Comments 1. The article is too long, which makes difficult to finish reading it. Maybe it should be divided in two parts (assuming that the specific comments listed above are addressed satisfactorily), one for the definition of water masses, and another for the discussion of the effects of the water masses on biogeochemical properties. This should also take care of the too long discussion section.

2. A substantial review of English grammar is needed; there are too many sentences that need revision as to be listed here.

3. line 122: Do you mean surface waters in the interior GoM?

---

## Author Comment (AC1) · 5 Nov 2019

We would like to thanks the reviewer for the positive criticism and recommendations. Below you find a point-by-point response to all comments.

C1 The attempt to eliminate overlaps between some of the water masses, and the introduction of the new surface water mass, is appreciated. However, the justification on the need to change the conventional ranges for the gulf's water masses is not convincing, and the way the water masses were defined is weak. Below, I am including

a number of scientific issues that need to be addressed for making a more convincing case, and for improving the presentation and readability of the manuscript. Specific Comments 1. The definition of water mass FISW does not satisfy conventional approaches to name the body of water and delineate its properties. a) Note that a water mass is defined as a body of water with a common formation history. Given that 'FISW' is under the continuous action of highly dynamic and variable forcing (wind stress, insolation, air-sea fluxes, precipitation, and mesoscale eddy dynamics), is it possible to attribute a common formation history to this body of water? b) Since the name of the water mass usually relates to its major area of residence, the name FISW is not appropriate for this potentially new water mass. c) A water mass is often found in regions well beyond its formation region. Is this condition satisfied in the case of FISW? I do not think so. This condition is difficult to be verified. d) A water mass can be identified away from its formation region because its elements retain its properties, in particular its potential temperature and salinity. Given that FISW extends over surfaces waters where irreversible vertical mixing is continuously changing the temperature of the body of water (diabatic turbulent process), temperature is not a conservative property over these surface body of water. Note that classical water masses that form in the surface usually sink into deeper waters away from surface forcing, which allows them to retain its original properties for long periods of time over grate distances. e) In order to accurately define a water mass, it is necessary to include information about its standard deviation; some water masses only require a single combination of T-S and its standard deviation, while delineating other water masses may require defining a T-S relationship and an envelope for the standard deviation. This requirement must be satisfied in the definition of FISW (no information is given about its standard deviation). Is its standard deviation small enough? Rejoinder: We are in complete agreement with the referee in that our newly identified Freshwater Influenced Surface Water (FISW) is not a water mass but a Water Type because it does not fit the description of a Water Mass in almost any sense (Tomczack, 1999; Emery, 2003). Other identifiable waters, termed in the MS as water masses, were so named following the older literature to minimize confusion.

(Morrison et al., 1983; Vidal et al., 1994; Rivas et al., 2005; Portela et al., 2018). Also, it is not the intent of this work to trace water masses to their core. Our interest is to better delineate the boundaries of the water masses in the Gulf of Mexico, insofar as possible. New physical and chemical parameters, both conservation and non-conservative, are added in the water mass concept (Tomczak, 1999). These additional variables exhibit different importance in defining a water masses but are complementary to each other and provide a more solid basis for the water mass definition.

C2 2. The definitions of water masses presented here need to be compared again historic definitions that are well established in the scientific community. Modifying table 2, and creating a new figure showing the different ranges of the water masses reported in the literature, will help in evaluating whether we need a new definition of the Gulf's water masses.

Rejoinder: With the exception of FISW, we have used the assigned water mass names from the literature (See references above). The referee's comment about using standard deviations of T and S (but largely T) to better characterize the water masses from our measurements and from data available from the literature is intriguing, but is not common practice (Emery, 2003). Water masses are generally typified by a range of T and S values (again, Emery, 2003). This is because the assignment of a volume from which T and S means and standard deviations can be computed is highly subjective, and adjoining water masses differ little in their physical properties. Because of this, an important component of our effort is the addition of chemical parameters to improve water mass identification, especially if there is overlap in the T-S diagrams. New Table 1 (attached file) shows the different ranges of the water masses reported in the literature vs. this work. Nevertheless, for the XIXIMI data, we calculated the mean relationship between temperature and salinity and the standard deviation of salinity as a function of temperature to choose the limits of the "natural" spread of the data and eliminate the outliers.

3. Why is the overlap between CSW and GCW a big deal? There is always a transition

region between water masses in every ocean (that is why we need a standard deviation in trying to isolate the dominant characteristics of the water mass). In order to justify the idea that we need to get rid of this overlap, this manuscript needs to quantify the error related to the conventional and new definitions, including a quantification of the effects on the density field. Are these errors significant, or are both at the noise level? If errors from both definitions are at the noise level, no new definition is needed.

Rejoinder: In the classification proposed by Portela et al. (2018) the overlap in the thermohaline ranges of the CSW and GCW was overlooked. However, we point out that by not including the full range of T & S values from 0 to 50 m, the actual volume of the CSW within the GoM would be underestimated, affect hydrography, and potentially, estimates of productivity. This effect is clearly illustrated in Figure 9a from Portela et al. (2018) that shows the loss of information that results when you overlap CSW and GCW. Figure 9b shows the new boundaries proposed in this work with no overlap. Figure 9b also shows that high oxygen values are principally a property of the GCW and not CSW.

The manuscript is missing an in-depth review of the-state-of-the-art on the formation of GCW. Rejoinder: We hae added the following paragraph to the MS: One of the mechanisms of formation of the Common Gulf Water (GCW) was that described by Elliott (1979, 1982), which states that the GCW is formed during the autumn and winter months as a result of the vertical convective mixing induced by the cold fronts (Nortes) that spread over the entire Gulf. On the other hand, Vidal et al. (1992, 1994) points out that the other mechanism of formation of the GCW ($\sigma\theta$ = 24.5 to 25.5 mg • cm3; S = 36.3 to 36.4; T $\sim$ 22.5 ° C) is the product of the release of the anticyclonic eddies coming from the CL in the northwestern region of the GoM, as well as in the winter when the wind regime produces a mixed layer of approximately 170 m that dilutes the SUW. Additionally, Vidal et al. (1992, 1994) mentions that the core of the GCW within the western region is heated by solar radiation and reaches minimum values of density $\sigma\theta$ = 25 mg • cm3 during the summer. On the other hand, Portella et al. (2018)

mentions that the GCW evidences the smoothing of the NASUW properties, which takes place throughout the lifespan of the LCE, mainly during the winter mixing.

4. It is not clear how the LC cycle can be used to eliminate the overlap between CSW and GCW since the LC does not transport GCW. This is an important issue in the approach presented here. Also note that Caribbean anticyclones can also make it into the Gulf transporting CSW. Moreover, atmospheric forcing could erase the CSW signature in winter over the Gulf. Some misleading statements regarding this issue are:

Rejoinder: An important result of our paper is that the DO marks the upper boundary of the GCW. The overlap, a mathematical construct, is not needed, nor needs to be explained.

a) This explanations in 455-459 and 553-554 are convoluted. Since the LC does not transport GCW (water mass originated in the GoM), and the CSW is presumably only found in the LC, how the CSW ends up on top of GCW? Something does not make sense here.

Rejoinder: The CSW is less dense than the GCW at all times largely caused by higher temperatures acquired in the Caribbean basin. In winter, the flow of CSW through the Yucatan Channel is a fraction of its summer flow, causing the retraction of the LC (Delgado et al., 2019 - In Press, and others) and its "signature" becomes dispersed in the GoM by "Nortes" winds as the CSW mixes with the SUW and the GCW.

b) Too much emphasize is put on the idea that the "weakening" of the LC is associated with the absence of CSW in the GoM (423). However, previous studies (cited in the present article) claim that the absence of CSW is becasue this water is continuously transformed in the GoM by wind forcing. Since the latter idea weakens the hypothesis that the LC cycle can be used in dealing with the overlap between CSW and GCW, this issue needs to be addressed in detail. Is CSW absent in winter in both the GoM and Caribbean Sea? If it is missing also over the Caribbean Sea, then atmospheric processes control the variability of this water mass over both the Gulf and the area of

formation.

Rejoinder: There is nothing in the literature about "disappearance" of the CSW in the Caribbean Sea (Corredor and Morel, 2001. JGR, 106:415-417). We have explained its absence in the GoM during winter in the paragraph above.

c) In lines 428-430 it is claimed that there is a salinity contribution to CSW in the GoM. Is not supposed that CSW acquires its distinctive high salinity values over the Caribbean Sea? A local addition of salinity within the GoM is against the conventional definition of water mass.

Rejoinder: No, the CSW acquires its low salinities (relative to its temperature) in the Caribbean basin, due to the influx of fresh water from the Amazon and Orinoco Rivers and from rain. (Corredor and Morel, 2001. JGR, 106:415-417). Inside the GoM the continuous winter mixing and subsequent restratification of the upper layer could be responsible for the increased salinity of the CSW in the GoM as compared to its Caribbean origin. Other possible explanation for this increase in salinity is thermal convective mixing due to the "Nortes" events, which would erode the salinity maximum of the NASUW during winter, with the subsequent salinity increase of the above layer. These two hypotheses are not exclusive, they both could act to increase the salinity of the CSW, and the predominance of one or another process would depend on the season of the eddy shedding (Portella et al. 2018). C3

5. I understand that water masses formed at the surface at higher latitudes retain its DO because they sink and move away from regions of intense atmospheric forcing. However, in the case of water masses that remain in the surface, is it valid to use DO for characterizing their properties? I am not sure about this, since intense vertical mixing acting over these bodies of water makes them diabatic (their properties are non-conservative). Note that DO is a function of temperature, and temperature is non-conservative in water parcels over the ocean mixed layer and upper thermocline. Also note that the LC cycle is not needed to have the variability in DO documented here

(562-564). It needs to be shown the variability in DO is not caused by atmospheric forcing in surface water masses; otherwise, it cannot be used in characterizing surface water masses.

Rejoinder: The referee is correct in pointing out that that the use of DO has pitfalls; one of them is its dependence on temperature and salinity even when conservative, as in deep waters, and the other is loss or gain across the atmospheric boundary. Nevertheless, we use it in the first 50 m because efflux and influx across the air sea boundary are relatively slow. An example of this "memory" is that the usual DO maximum lies at about 10 m and not at 0 m. Nevertheless, in this work we support our conclusions by calculating AOU over the same DO range. Now, AOU is both temperature and salinity independent, and clearly shows the boundary between net surface production and the net respiration below it.

6. An important analysis and methodology for redefining the water mases are given in Fig. 4 and appendix A. These approaches can be significantly simplified by satisfying the conditions listed in item 1 above; using the standard deviation can be particularly helpful. Rejoinder: As we pointed out earlier, away from their core, T & S differences between adjoining water masses are small while differences between biogeochemical variables tend to be larger and independent of temperature. Calculated differences between means are made less significant by the subjective estimation of the volumes considered. Calculations of the standard deviations of water masses are not common.

7. a) What are the source of nitrite and DIC contained in FISW?. Is the seasonal variability in these properties related to vertical mixing (and cooling of the sea surface), since these two chemicals depend on temperature? Because these properties reflect the dynamic and variable characteristics of surface waters (409-412). Can they be used in delineating water masses? They are clearly impacted by the seasonal cycle of insolation and vertical mixing over the upper ocean, and likely also by local biogeochemical processes. Rejoinder: No, nitrate (combined with nitrite) and DIC are in units of mass/mass, thus temperature and salinity independent. Their sources are variable

freshwater and mixing, and we cannot speculate on them. Commonly, they are seen as near conservative properties in the water column, and that changes in their vertical concentrations are caused by the processes of respiration and phosynthesis. Thus, nitrate is non-volatile and reflects them, being very low at the surface when mixing is slow, and somewhat higher when mixing is greater. The same can be said for DIC whose air-sea exchange rate is about 1/5 that of DO. So, while they are only quasi conservative at the surface, below the surface, they are useful for characterizing water masses (think of the oxygen minimum in Tropical Atlantic Central Water). At the surface, the shape of the profiles really has to do with the relative rates of mixing vs. photosynthesis and respiration. If mixing processes dominated, then all profiles would look like straight vertical lines. When this is not the case, biogeochemical processes dominate. How to separate the two? Using salinity as a very conservative property and "normalizing" the other variables to it (Broenkow, W., Limnology and Oceanography, 10:40-52, 1965).

8. Where is the analysis of the Brunt-Vaisala frequency (345-247) being shown? Rather than buoyancy alone, it is the criticality of the Richardson number (Ri<1/4) that is used to identify periods of vertical mixing. In addition to the buoyancy frequency, measurements of horizontal current vertical shear are also needed in the computation of Ri.

Rejoinder: Brunt-Väisälä frequency was calculated to estimate the stability of the water via TEOS-10 (See Fig. a and b in the attach file). The theme of the mixed layer is of great interest in the GoM for primary production studies, but there are several difficulties. In the GoM, mixed layer follows a clear seasonal cycle characterized by a deepening in winter. Under these conditions, we considered useful estimate the stability of the water by Brunt-Väisälä. The figure shows the vertical profiles of potential density anomaly and the Brunt-Vaisala stability parameter during winter (a and b) and summer (c and d). In summer, with the presence of the CSW, the mixed layer lies above the nutricline, as determined by the upper reaches of the GCW (Fig. c y d).

As mentioned throughout this work and by observations by other authors (Delgado et., 2019), during the summer months with the entry of LC, the presence of oligotrophic waters dominates in the first 100 m. On the contrary, in winter, in the absence of Caribbean water and greater vertical mixing induced by the Nortes, will favor a deep mixed layer (See Fig. a and b in the attach file). We did have all the data needed to calculate the Richardson number and we feel useful enough estimate the stability of the water by Brunt-Väisälä (we can include the figure in the text if needed).

9. Another possibility for explaining the seasonal change in the nutricline and carbocline (387-390), is that these properties are a function of the seasonal cycle of the wind stress and insolation since these properties clearly are a function of temperature as per Fig. 8.

Rejoinder: As explained in Rejoinder 7, Nitrate and DIC concentrations are independent of temperature and salinity. What shows in Fig. 8 is the combined result of mixing and in-situ processes. Because mixing affects all variables, the concentrations of Nitrate and DIC appear to be functions of temperature. This is particularly true in surface waters where wind mixing is rapid.

10. The vertical exchange of chemical properties between water masses discussed in 484-486 can occur by diffusion (very low time scale), or by diapycnal mixing that requires vertical mixing and water mass transformation. What is the more likely mechanism for explaining this conundrum? Again, the introduction is needs an in-depth discussion on the formation of GCW.

Rejoinder: We don't see that there is a problem here. Even at 50 m diffusion, even eddy diffusion is too slow mix the two water masses in question. As pointed out earlier, the influx of CSW into the gulf via the LC is greatly reduced in winter, and what there is mixed into the GCW and SUW by the northerly winds. We also discussed the formation of GCW in depth in our answer for question 3.

11. Note that the Mississippi River plume (508-510) also extends southward into the

LC and associated eddy field; this plume can also leave the GoM through the Florida Straits. This topic needs a review of the state-of-the-art, since river runoff can be an important contribution to FISW.

Rejoinder: Yes indeed, the Mississippi River plume can be found in the south and there are other rivers along the Mexican coast to be considered. In the MS we stated that the low salinity characteristics of FISW are largely the result of the fresh water that enters the GoM.

Technical Comments 1. The article is too long, which makes difficult to finish reading it. Maybe it should be divided in two parts (assuming that the specific comments listed above are addressed satisfactorily), one for the definition of water masses, and another for the discussion of the effects of the water masses on biogeochemical properties. This should also take care of the too long discussion section. 2. A substantial review of English grammar is needed; there are too many sentences that need revision as to be listed here. Thanks, we will do that.

3. line 122: Do you mean surface waters in the interior GoM? Yes.

Please also note the supplement to this comment:
https://www.biogeosciences-discuss.net/bg-2019-340/bg-2019-340-AC1-supplement.pdf

---

## Referee Comment (RC2) · Anonymous Referee #2 · 11 Nov 2019

In the manuscript "A New Characterization of the Upper Waters of the Central Gulf of Mexico based on Water Mass Hydrographic and Biogeochemical Characteristics" the authors present a new classification of water masses based on data collected in 5 cruises over a period of 6 years. This is an interesting exercise in water mass classification, using a valuable data set that spans different seasons and years, and includes physical and biogeochemical data.

Before addressing the substance of the paper, I have two recommendations for subsequent revisions:

[Figure]

- The use of English needs to be improved substantially. Some sentences were quite honestly difficult to understand at all, others lacked a subject, or were grammatically incorrect. As an example, the authors keep using "in this manuscript" as if it were a subject. I am surprised that none of the co-authors took the time to read and improve the grammar.

- Editing: this reads more as a dissertation/report chapter than a review paper. As an example, do three lines of text merit a whole sub-section (line 226-229)? Do we really need 3 subsections (e.g. 2.2.1, 4.1.2 etc)? On top of addressing content concerns, the authors need to edit the manuscript substantially for style.

Content-related concerns. Major issues:

- The authors' statement that data is not available at this time, when some data is from 2010, makes me think there is simply no plan to make them available at all. Given BG's policy regarding data sharing, I find this problematic. I leave it to the editors to evaluate whether papers can be published on BG without releasing the data used to sustain their conclusions.

- The authors need to make a better job of justifying the need for a new classification of water masses and how this work resolves issues that previous classifications could not address otherwise. What were those issues and why couldn't other classifications work? I suggest a table that summarizes the water masses proposed by previous authors to help in the comparison. In particular, the authors rely heavily on a paper by Portela et al. Those should definitely be referenced more clearly here.

- I'm not convinced about the definition of FISW, a water mass whose salinity changes depending on river runoff and precipitation, and time of year. Does it really qualify as a water mass? How far from the area of formation can it be found other that due to eddy transport? For how long/far does it maintain the same characteristics? Another reviewer mentioned this in their comments and I fully agree with their opinion.

[Figure]

- The authors mention initially that their data is collected from the Mexican side of the GoM. I would like to see a justification for how they extended their results to waters on the US side, or otherwise clarify throughout the manuscript that this applies only to the area covered during their cruises (i.e. the Mexican section of the GoM).

- The authors' new classification is based on T, S, DO. I'd like to see clarification on how NO3 and DIC add additional value to the definition of the water masses. Otherwise, I would recommend that the authors streamline this paper, focusing on T, S, DO and the definition of the water masses, and save the DIC, NO3 discussion for a separate work. DIC in particular did not seem to add anything relevant.

Small comments: - I suggest adding a table that summarizes the five cruises and that lists years, seasons, etc. as it will be useful to reference back to it during the discussion of the results.

- Every instance of "in this manuscript/in this work" immediately followed by a verb needs to be changed to "this manuscript/this work" or "in this manuscript/work we" etc.

- The affiliations for the authors should be in order of appearance, e.g. for Jose Martin Hernandez-Ayon the affiliations should be numbers 1,2, not 1,5.

- On multiple occasions there is an "H" preceding a number. Why? This does not seem to be related to the parameter (e.g. sometimes potential density values are preceded by H, sometimes they aren't. Likewise for temperature or DO).

- Line 52: the windy "nortes" season. Are these northerly winds? Please explain for those unfamiliar.

- Several times throughout the text: do not use "approx.", write the full word.

- Line 117: consider rephrasing (and improve English). DO shows high variability. Do you mean a range more than 200 umol/kg? This is not possible based on the legend in figure 1b (this scale shows around 125 umol/kg between the minimum and maximum). Is the scale incorrect?
Line 178-180: this line is difficult to understand.

Line 213-214: This so-called Gulf Common Water. This actually means "this supposed Gulf Common Water". So is it GCW or not? If it is, then do not use "so-called".

Lines 226, 231: the text jumps from section 2.2.4 to 2.4. Where is section 2.3? Again, thoroughly revise text to make it more article style and less dissertation style.

Line 273: Isn't this section 3, not 2?

Line 335: why does it need to be better defined? Please elaborate further.

Lines 345-347: Hard to understand sentence. Why not describe the results of the frequency analysis a bit more and why not show the figure?

Section 3.4: I suggest eliminating this section and focusing on the core results, i.e. the classifications of the water masses. Otherwise, the authors need to better argue for the added value that these parameters bring to help define the water masses.

Line 379: There has been no mention of the TACW since the introduction. In lines 274-278 it was not listed as a relevant pattern. The TACW is only mentioned here. How relevant is it overall?

Line 402: Is this for the central and western GoM in general or is this the central and western Mexican GoM?

Line 423: This is a concluding remark that is not supported by the preceding paragraphs. Either, move this sentence further down in the text to after the next couple of paragraphs, or simply delete it.

Lines 526-533: move this to the conclusions or remove altogether to shorten length of manuscript.

Line 539: "reaching down to 90 m in spite." In spite of what? This sentence does not make sense.

Lines 787-790: this reference corresponds to a doctoral thesis written in French 15 years ago. Was there no publication in a peer reviewed journal ever published? Is this reference available to the general public and is there no other reference that would be more adequate and readily available?

---

## Author Comment (AC2) · 17 Nov 2019

1- The use of English needs to be improved substantially. Some sentences were quite honestly difficult to understand at all, others lacked a subject, or were grammatically incorrect. As an example, the authors keep using "in this manuscript" as if it were a subject. I am surprised that none of the co-authors took the time to read and improve the grammar. Rejoinder: We are sorry about the errors. The MS was rushed and submitted before a thorough review of the English version was made by all authors. Although none of us is a native English speaker, we feel that the MS is now grammat-

ically acceptable. We also have made many changes, largely deletions, to produce a better focused article. Although past work has been reviewed, the article is not a review paper, but a presentation of new work and of the insights provided by it.

2- Editing: this reads more as a dissertation/report chapter than a review paper. As an example, do three lines of text merit a whole sub-section (line 226-229)? Do we really need 3 subsections (e.g. 2.2.1, 4.1.2 etc)? On top of addressing content concerns, the authors need to edit the manuscript substantially for style. Rejoinder: We have removed the three subsections 2.2.1 to 2.2.4 and renumbered the the text. Content-related concerns. Major issues:

3- The authors statement that data is not available at this time, when some data is from 2010, makes me think there is simply no plan to make them available at all. Given BG's policy regarding data sharing, I find this problematic. I leave it to the editors to evaluate whether papers can be published on BG without releasing the data used to sustain their conclusions. Rejoinder: We are releasing all data in accordance with the publisher's guidelines.

4- The authors need to make a better job of justifying the need for a new classification of water masses and how this work resolves issues that previous classifications could not address otherwise. What were those issues and why couldn't other classifications work? I suggest a table that summarizes the water masses proposed by previous authors to help in the comparison. In particular, the authors rely heavily on a paper by Portela et al. Those should definitely be referenced more clearly here. Rejoinder: We think that we have made clear that this work is focused on offshore surface waters of the GoM, waters previously neglected in the literature. Older classifications did not address them. Therefore, it is not evident to us why an additional table is needed. We have mentioned the contributions of Portela et al., 2018, seventeen times in the MS, and in addition, Esther Portela was an early reviewer of this work, a fact that we gratefully acknowledged.

5- I'm not convinced about the definition of FISW, a water mass whose salinity changes depending on river runoff and precipitation, and time of year. Does it really qualify as a water mass? How far from the area of formation can it be found other that due to eddy transport? For how long/far does it maintain the same characteristics? Another reviewer mentioned this in their comments and I fully agree with their opinion. Rejoinder: We are in complete agreement with both referees, and have addressed this issue by considering FISW a water type rather than a water mass. Please refer to our rejoinder to Referee1.

6- The authors mention initially that their data is collected from the Mexican side of the GoM. I would like to see a justification for how they extended their results to waters on the US side, or otherwise clarify throughout the manuscript that this applies only to the area covered during their cruises (i.e. the Mexican section of the GoM). Rejoinder: Throughout the MS, we have mentioned that our results pertain to the XIXIMI grid that covers the southern GoM (Fig. 1). Nevertheless, there is no clear oceanographic boundary between the northern and southern GoM. Although our measurements have been limited to the southern GoM, our results for the southern GoM have also been supported by our analysis of CARS2009-archived data; a world-wide data base that includes the entire GoM. This suggests that our observations may carry to the north as well. Additionally, Portella et al., (2018) classified water masses west of 88oW e.g., much of the entire western GoM, north and south, using 15,854 profiles with temperature, salinity, and DO obtained from the World Ocean Database, 2013 (WOD13), 14 research cruises between 2010 and 2016, and six missions performed continuously between May 2016 and August 2017 by a fleet of gliders of the Grupo de Monitoreo OceanograÌfico con Gliders (GMOG) mainly from the US side, and 17,695 profiles with information on only temperature and salinity, including data from recent ARGO observations. Our data from the six XIXIMI cruises match with the water masses of the Portella et al., (2018) classification rather well (Fig.9). C2

7- The authors' new classification is based on T, S, DO. I'd like to see clarification on

how NO3 and DIC add additional value to the definition of the water masses. Otherwise, I would recommend that the authors streamline this paper, focusing on T, S, DO and the definition of the water masses, and save the DIC, NO3 discussion for a separate work. DIC in particular did not seem to add anything relevant. Rejoinder: We do not wish to remove the biogeochemical variables from this MS. While our initial classification was, as is customary in physical oceanography, based on T, S, and DO, it also was supported by nitrate, AOU, DIC, and mention of these relationships is made throughout the MS. Our work is probably one of the first to merge these data with the hydrography of the GoM. While the physical framework supports the GoM, its effect on the biology/chemistry is, arguably the most important from the ecological point of view. We are aware that this work discusses the interrelationships among the biological/chemical variables only briefly, but it points out that the interactions among physical and biogeochemical variables are linked, and that this synergy will ultimately provide a better understanding of oceanic productivity and other collateral issues important to mankind.

Small comments: 8- I suggest adding a table that summarizes the five cruises and that lists years, seasons, etc. as it will be useful to reference back to it during the discussion of the results. Rejoinder: This information is already available in the "Methods" section. An additional table does not seem to be justified.

9- Every instance of "in this manuscript/in this work" immediately followed by a verb needs to be changed to "this manuscript/this work" or "in this manuscript/work we" etc. Rejoinder: We appreciate your recommendation for writing correction and fluency, and have eliminated virtually all of "in this manuscript/in this work".

10- The affiliations for the authors should be in order of appearance, e.g. for Jose Martin Hernandez-Ayón the affiliations should be numbers 1,2, not 1,5. Rejoinder: Thank you. We have listed the author's affiliations in order of appearance.

11 - On multiple occasions there is an "H" preceding a number. Why? This does

not seem to be related to the parameter (e.g. sometimes potential density values are preceded by H, sometimes they aren't. Likewise for temperature or DO). Rejoinder: This computer-generated error was corrected.

12- Line 52: the windy "nortes" season. Are these northerly winds? Please explain for those unfamiliar. Rejoinder: In the Gulf of Mexico, the cold winter fronts move in a southeastern direction carried by strong winds from the north - northwest, hence they are known as "Nortes" winds. Usually, these cold fronts form on land and enter the gulf waters producing an intense interaction between the dry and cold polar air mass as they advance over the warm gulf waters. Likewise, when moving from land to ocean, due to the change in surface roughness, the winds that accompany these fronts tend to accelerate.

13- Line 117: consider rephrasing (and improve English). DO shows high variability. Do you mean a range more than 200 umol/kg? This is not possible based on the legend in figure 1b (this scale shows around 125 umol/kg between the minimum and maximum). Is the scale incorrect? Rejoinder: This is correct, DO variability was about 125 umol/kg-1. C3

14-Line 178-180: this line is difficult to understand. Rejoinder: We rewrote it as follows: "For the initial water mass identification we first used the limits described by Vidal et al. (1994), Morrison et al. (1983), Nowlin et al. (2001), and the recent classification proposed by Portela et al. (2018), as shown in figure 1a"

15-Line 213-214: This so-called Gulf Common Water. This actually means "this supposed Gulf Common Water". So is it GCW or not? If it is, then do not use "so-called". Rejoinder: Thank you. The line was rewritten as "we called this Gulf Common Water (GCW)"

16-Lines 226, 231: the text jumps from section 2.2.4 to 2.4. Where is section 2.3? Again, thoroughly revise text to make it more article style and less dissertation style. Rejoinder: The error was corrected.

17-Line 273: Isn't this section 3, not 2?

Rejoinder: The error was corrected.

18-Line 335: why does it need to be better defined? Please elaborate further. Rejoinder: We removed this line from the "Result" section and we elaborated further in the "Discussion" section (see Discussion, sections 4.1).

19-Lines 345-347: Hard to understand sentence. Why not describe the results of the frequency analysis a bit more and why not show the figure? Rejoinder: We will include the figure showing the seasonal differences in the Brundt-Vaisala oscillations in the supplementary material as it was one of the first reviewer's suggestions. A brief description of the figure will be included in the manuscript.

20-Section 3.4: I suggest eliminating this section and focusing on the core results, i.e. the classifications of the water masses. Otherwise, the authors need to better argue for the added value that these parameters bring to help define the water masses. Rejoinder: See our answer from question 7.

21-Line 379: There has been no mention of the TACW since the introduction. In lines 274-278 it was not listed as a relevant pattern. The TACW is only mentioned here. How relevant is it overall? Rejoinder: We considered TACW relevant mainly during winter, when the CSW is absent. We also observed that TACW is shallower than in summer. The proximity of the TACW to the GCW facilitates the vertical exchange of chemical properties towards the surface. Convective mixing leads to low DO concentrations of the TACW to be reflected in the GCW, as well as causing an observable increase in nitrate and DIC concentrations (this is part of the discussion section 4.1.2).

22-Line 402: Is this for the central and western GoM in general or is this the central and western Mexican GoM? Rejoinder: We going to rewrite this line as: A recent detailed analysis all GoM including the US side by Portela et al. (2018)

23-Line 423: This is a concluding remark that is not supported by the preceding paragraphs. Either, move this sentence further down in the text to after the next couple of paragraphs, or simply delete it. Rejoinder: We rewrote as follow: "Here, we emphasize that the seasonal "pulsing" of the LC and the Yucatán Current into the GoM explains the presence of CSW in summer and we attribute its absence to the weakening of the LC in winter"

24-Lines 526-533: move this to the conclusions or remove altogether to shorten length of manuscript. Rejoinder: We have moved those lines to the "Conclusion".

25-Line 539: "reaching down to 90 m in spite." In spite of what? This sentence does not make sense. Rejoinder: We rewrote this as follow: "One of the biological implications of the presence of CSW is that it is oligotrophic and is found down to 90 m" C4

26-Lines 787-790: this reference corresponds to a doctoral thesis written in French 15 years ago. Was there no publication in a peer reviewed journal ever published? Is this reference available to the general public and is there no other reference that would be more adequate and readily available? Rejoinder: We removed this reference from the list.
* * *